

# Local transformations of multiple multipartite states

**Antoine Neven, David Gunn, Martin Hebenstreit and Barbara Kraus**

Institute for Theoretical Physics, University of Innsbruck, A–6020 Innsbruck, Austria

## Abstract

Understanding multipartite entanglement is vital, as it underpins a wide range of phenomena across physics. The study of transformations of states via Local Operations assisted by Classical Communication (LOCC) allows one to quantitatively analyse entanglement, as it induces a partial order in the Hilbert space. However, it has been shown that, for systems with fixed local dimensions, this order is generically trivial, which prevents relating multipartite states to each other with respect to any entanglement measure. In order to obtain a non-trivial partial ordering, we study a physically motivated extension of LOCC: multi-state LOCC. Here, one considers simultaneous LOCC transformations acting on a finite number of entangled pure states. We study both multipartite and bipartite multi-state transformations. In the multipartite case, we demonstrate that one can change the stochastic LOCC (SLOCC) class of the individual initial states by only applying Local Unitaries (LUs). We show that, by transferring entanglement from one state to the other, one can perform state conversions not possible in the single copy case; provide examples of multipartite entanglement catalysis; and demonstrate improved probabilistic protocols. In the bipartite case, we identify numerous non-trivial LU transformations and show that the source entanglement is not additive. These results demonstrate that multi-state LOCC has a much richer landscape than single-state LOCC.


doi:10.21468/SciPostPhys.11.2.042

# 1 Introduction

Multipartite entanglement is a central phenomenon across quantum theory, underpinning large swathes of physical phenomena. In condensed matter physics, entanglement characteristics of many-body systems can be utilized to study phase transitions [1] and to derive numerical algorithms using tensor network states [2–4]. Within quantum information theory, entanglement is considered to be the resource which allows quantum technologies to outperform their classical counterparts. That is, having access to an entangled state enables quantum information-processing tasks that cannot be achieved classically, such as teleportation [5], measurement-based quantum computation [6] and entanglement-based quantum communication [7, 8]. Despite its importance, we are still far from a complete understanding of entanglement. Any new insight into this intriguing property of quantum systems will provide deeper understanding of its relevant applications and advance the fields related to it.

The predominant feature of entanglement is that it cannot be created locally. For this reason, entanglement is often studied in the physical framework of the "distant labs" model, in which individual labs, which share a multipartite state, are spatially separated and constrained to apply Local quantum Operations, possibly assisted by Classical Communication (LOCC). As entanglement cannot be created or enhanced using LOCC, if a state can be transformed into another via LOCC, it has to be at least as entangled as the final state. As a consequence, LOCC induces a partial order on the Hilbert space and any entanglement measure, i.e. any function quantifying the entanglement resource of states, has to be non-increasing under LOCC [9]. This order is only partial as there exist pairs of states which are incomparable under LOCC, i.e. neither can reach the other via LOCC. States which can be generated locally (if no superselection rules or the like are imposed [10]) can be described as a convex combination of product states and are called separable states. Hence, in the resource theory of entanglement, the free states are separable states and the free operations are precisely LOCC [9, 11–13].

The characterization of pure-state entanglement was particularly successful in bipartite systems, for which Nielsen's celebrated majorization criterion [14] gives a necessary and suf-

ficient condition for the existence of LOCC transformations between pure states. Moreover, as a direct consequence of Nielsen's criterion, there exists (up to local unitaries) only one maximally entangled state, from which the whole Hilbert space is accessible via LOCC. Furthermore, although entangled states that are not maximally entangled cannot be deterministically transformed into the maximally entangled state of the Hilbert space, such a transformation is always possible via a Stochastic LOCC (SLOCC) protocol, i.e. an LOCC protocol with non-vanishing probability of success. Therefore, all entangled bipartite states (with the same local ranks) form a single SLOCC equivalence class [15]. Finally, in the asymptotic limit, copies of a bipartite state can be deterministically and reversibly converted into maximally entangled states at a rate given by the Von Neumann entropy of the reduced state [16,17]. Consequently, both in the single copy regime and in the asymptotic limit, one can study pure-state bipartite entanglement through maximally entangled states. The intermediate regime of a finite number of copies was studied [18] and an optimal protocol for entanglement concentration was provided [19].

Although LOCC has, by definition, a very complicated mathematical structure [12, 20], with a possibly unbounded number of rounds of communication between the parties, bipartite LOCC protocols can always be reduced to simple one-round protocols [21]. This is not the case for multipartite LOCC, for which it has been shown that certain LOCC protocols require an unbounded number of communication rounds [22]. Similarly, even though most known multipartite LOCC transformations are all-deterministic (i.e. do not need any probabilistic intermediate steps) [23,24], it was shown that some LOCC transformations cannot be achieved without probabilistic steps [25]. This results in multipartite LOCC being much more complicated to characterize than bipartite LOCC [12, 20].

Even in three-qubit systems, there are considerable differences to bipartite systems. There exist two distinct SLOCC classes of fully-entangled three-qubit states [15]. This means that there are two different (and incomparable) types of entanglement for three-qubit states, in contrast to the single type of bipartite entanglement. This also implies that there does not exist a single maximally entangled state of three qubits. The maximally entangled state of bipartite systems can be generalized into a set [26], called the Maximally Entangled Set (MES), containing the minimal number of states required to reach the whole Hilbert space via LOCC transformations. Though of zero measure in the Hilbert space of three-qubit states, this set nevertheless contains an infinite number of states [26].

The problem only worsens with larger system sizes and/or higher dimensions. First, there is generically an infinite number of SLOCC classes [27]. Second, for homogeneous systems (i.e. multipartite systems with subsystems of equal dimension) of at least four parties, almost all pure states are isolated under LOCC [26, 28–30]. That is, almost all pure states can neither be reached from, nor transformed into, any other pure state via LOCC. As a consequence, the partial order induced by LOCC is generically trivial and the MES is of full measure in the Hilbert space. These results show that, given an arbitrary multipartite state (from a homogeneous system), it is generically impossible to find another state which is less entangled with respect to all entanglement measures. Moreover, the optimal resource, i.e. the MES, has full measure in the considered Hilbert space.

Nonetheless, the identification of the optimal resource is crucial in recognizing new applications of multipartite entanglement. In this sentiment, various approaches have been pursued to identify an optimal resource, and both mathematically [31–34] and physically [35–38] motivated extensions of LOCC have been considered. Here, we focus on a physical extension, which is to characterize LOCC in the multi-state (non-asymptotic) setting. This setting is indeed very practical as, assuming the parties have access to a quantum memory, it amounts to several labs trying to combine the resources of several shared states by acting simultaneously (though still locally) on them. Even though one would, in such a practical setting, inevitably

have to deal with mixed states, we focus here on pure state transformations for two reasons. First, from a theoretical point of view, understanding pure state transformations is a necessary step towards the study of mixed state transformations. Second, for experimental states that are "close" enough to pure states, the pure state transformations hold up to a certain fidelity of the final states. Consider an impossible transformation $|\psi\rangle \nrightarrow_{LOCC} |\bar{\psi}\rangle$, with $|\psi\rangle, |\bar{\psi}\rangle$ elements of the same Hilbert space, $\mathcal{H}$. Appending an auxiliary state $|\phi\rangle$ (which does not necessarily belong to $\mathcal{H}$), as an additional resource, may enable the transformation $|\psi\rangle \otimes |\phi\rangle \rightarrow_{LOCC} |\bar{\psi}\rangle \otimes |\bar{\phi}\rangle$ for some state $|\bar{\phi}\rangle$ (see Fig. 1). Multi-state LOCC transformations have been studied in various contexts, such as in catalytic transformations, where the auxiliary state must be preserved through the transformation [39]. Diverting from deterministic transformations, also SLOCC catalysis has been investigated in [40].

## MULTI-STATE LOCC

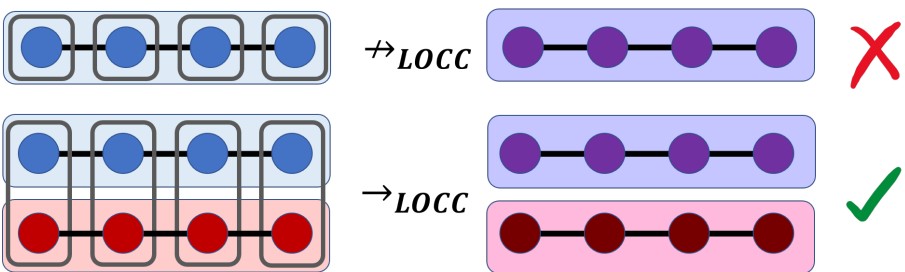

Figure 1: Illustration of an LOCC transformation between the states $|\psi\rangle$ and $|\phi\rangle$ that is not possible in the single-state regime, but that becomes possible in the multi-state regime, by adding the auxiliary state, $|\bar{\psi}\rangle$. Locality is now defined to include parts of both states. Note that transformations of the target state under LOCC are clearly a subset of transformations of the target state under multi-state LOCC.

There are several remarks in order. First, if new transformations from $k$ copies of a state to $k$ copies of another state can indeed be achieved, one could sort the entanglement contained in the states according to the ordering achieved in this specific multi-state setting, which could now be non-trivial. Second, note that this new order depends on the dimension of the Hilbert space to which the auxiliary state belongs. Note further that, even though transformations in the higher dimensional Hilbert space will generically still not be possible (for homogeneous systems), this does not imply that a multi-state transformation is generically not possible. The reason for this is that the multi-states are of measure zero in the whole Hilbert space. Finally, multi-state transformations could allow one to reach a state inside the MES from states outside the MES, which could imply that the multi-state equivalent of the MES is strictly smaller than the MES. Characterizing such a multi-state MES could be a step towards its reversible asymptotic version, called the Minimal Reversible Entanglement Generating Set (MREGS), introduced and studied in Ref. [41].

In this paper, we investigate LOCC multi-state transformations, both in the multipartite and bipartite settings. In both cases, we focus on two-state LOCC and investigate new transformations that arise in this setting. We show that, already in that case, the multi-state regime provides a much richer landscape of LOCC transformations than the single-state regime.

For multipartite states, we illustrate how much more powerful LOCC is in the multi-state regime by describing important new types of transformations this regime enables. For instance, even if the overall tensor products of the initial and final states are in the same SLOCC class, we show that a multi-state transformation can deterministically change the SLOCC class of the individual states with only Local Unitaries (LUs). In light of this possibility, it appears that one

has to consider, as potential final states, states belonging to a different SLOCC class (which generically means an infinite number of possibilities). As a consequence, the tools used to study LOCC transformations in the single-state regime cannot be used in the multi-state case. This suggests that characterizing all possible multi-state transformations is a formidable challenge. We nevertheless demonstrate important new features of multi-state transformations. For instance, we show that a state from the MES can be reached from two copies of a state outside the MES and that multipartite catalytic transformations can be achieved. In the event a multi-state transformation cannot be performed deterministically, we demonstrate that the maximum success probability of a joint multi-state transformation can be greater than the probability of transforming both states independently. Furthermore, we show it can be greater than even the maximum probability of either single-state transformation. Therefore, the multi-state regime also provides an advantage in probabilistic settings.

Regarding bipartite states, they have the big advantage that their entanglement can be studied through Schmidt coefficients, which naturally extends to the multi-state regime. Since Nielsen's majorization criterion also extends to the characterization of multi-state LOCC transformations of bipartite states, one could think that such transformations are simple to characterize. However, the difficulty stems from sorting the products of Schmidt coefficients that one gets in the multi-state regime (which is necessary for verifying the majorization condition). Such transformations have only been characterized assuming extra constraints, such as considering catalytic transformations [39, 42, 43]. In order to start systematically investigating bipartite entanglement in the multi-state regime, we focus here on LU transformations. Multi-state LU transformations have also been studied in the context of entanglement embezzlement [44]. In this paper, we give a full characterization of all possible transformations of a 2-qubit state (using an auxiliary state of arbitrary dimension) under LUs acting on the two states. This result then allows us to show that such LUs provide non-trivial transformations in almost all pairs of bipartite systems. Using then some of these non-trivial transformations, we demonstrate that the source entanglement [45] is a non-additive entanglement measure.

The remainder of the paper is structured as follows. In Section 2, we set up mathematical notations for the rest of the paper and review known results regarding multipartite LOCC transformations. In Section 3, we set the stage for our investigations of multi-state transformations. Section 4 is dedicated to multi-state multipartite LOCC transformations. We then consider bipartite multi-state transformations in Section 5. We finally draw conclusions in Section 6.

## 2 Preliminaries

In this section, we set out our notations and introduce mathematical tools that we will use throughout this paper. We also recall some important previous results about the characterization of LOCC state transformations. We consider multipartite states from the Hilbert space, $\mathcal{H} \cong \left(\mathbb{C}^d\right)^{\otimes n}$, i.e. $n$-partite states with local dimension $d$. Two states $|\psi\rangle$ and $|\phi\rangle$ are in the same SLOCC (resp. LU) class if they can be inter-converted via an SLOCC (LU) protocol. Stated mathematically, this is the case if and only if there exists a set of invertible operators $\{g^{(i)} \in \mathrm{GL}(d, \mathbb{C})\}_{i=1}^n$ (resp. unitary operators $\{g^{(i)} \in \mathrm{U}(d, \mathbb{C})\}_{i=1}^n$), such that $\otimes_{i=1}^n g^{(i)} |\psi\rangle = |\phi\rangle$ [15]. Throughout this paper, we will use superscripts to denote the subsystem local operators act on.

Not all SLOCC equivalence classes possess the same properties; they are classified in three different orbit types [46–49]. An SLOCC class is called *polystable* if it contains a critical state, i.e. a state for which all the single-party reduced density operators are maximally mixed. Critical states, such as the 3-qubit GHZ state, $|\mathrm{GHZ}\rangle \equiv \frac{1}{\sqrt{2}}(|000\rangle + |111\rangle)$, can be regarded

as highly entangled in the sense that they maximize many entanglement monotones [49]. In contrast, an SLOCC class from the *null-cone* contains entangled states, such as the 3-qubit W state, $|W\rangle \equiv \frac{1}{\sqrt{3}}(|001\rangle + |010\rangle + |100\rangle)$, for which the aforementioned monotones vanish. The last type of SLOCC class corresponds to classes which contain the so called *strictly semistable* states (see Appendix A for more details). Because LU operations are a trivial kind of LOCC, that can be applied to any state, we consider LOCC transformations that consist only of LUs as trivial transformations and ignore them in the following. Stated differently, we study LOCC transformations between LU equivalence classes of states. We restrict ourselves to studying transformations between fully-entangled states. That is, states for which all the single-party reduced density matrices $\rho_i$ have full rank, i.e. for which $\text{rank}(\rho_i) = d$, $\forall i \in \{1, \ldots, n\}$.

As will become apparent below, the local symmetries of states are central in the study of possible LOCC transformations. For each SLOCC class, we choose a representative state $|\psi\rangle$, called a seed state, and relate all the other states of the SLOCC class to $|\psi\rangle$ via local invertible operators. Given a seed state $|\psi\rangle$, we define the stabilizer of $|\psi\rangle$, $\mathcal{S}_\psi$, as the set of local invertible matrices that leave $|\psi\rangle$ invariant, i.e.

$$\mathcal{S}_\psi = \left\{ S = \bigotimes_{i=1}^{n} S^{(i)} \in \text{GL}(d, \mathbb{C})^{\otimes n} : S |\psi\rangle = |\psi\rangle \right\}. \tag{1}$$

Note that this set is not necessarily finite and that, from the stabilizer of a state, it is easy to determine the stabilizer of all SLOCC-equivalent states. We also define the set $\mathcal{N}_\psi$ as the set of local singular matrices which annihilate $|\psi\rangle$, i.e.

$$\mathcal{N}_\psi = \left\{ N = \bigotimes_{i=1}^{n} N^{(i)} \in \text{Mat}(d, \mathbb{C})^{\otimes n} : N |\psi\rangle = 0 \right\}. \tag{2}$$

As explained in the introduction, multipartite LOCC has a complex mathematical structure. However, LOCC is a (strict [50–52]) subset of the mathematically considerably more tractable class of Separable maps (SEP). A linear, completely positive, trace preserving map from the set of bounded linear operators acting on $\mathcal{H}$ to itself, $\Lambda : \mathcal{B}(\mathcal{H}) \to \mathcal{B}(\mathcal{H})$, is in SEP if it admits a Kraus decomposition in which all Kraus operators are separable [53]. That is, $\Lambda$ is in SEP if $\Lambda(X) = \sum_{i=1}^{m} M_i X M_i^\dagger$ for some $\{M_i\}_{i=1}^{m} \subset \mathcal{B}(\mathcal{H})$ such that $\sum_{i=1}^{m} M_i^\dagger M_i = \mathbb{1}$ and for all $i$, $M_i = \otimes_{j=1}^{n} M_i^{(j)}$ for some $\{M_i^{(j)}\}_{j=1}^{n}$. It is important to note that, in contrast to LOCC transformations, SEP maps do not have a physical interpretation, as it has been realized that not all SEP maps may be implemented through an LOCC protocol. Nonetheless, as an LOCC transformation is also a SEP transformation, we can use SEP transformations as a superset of the physical LOCC transformations.

In Refs. [29, 53], it was shown that, when restricted to transformations between fully-entangled pure states, a state $g |\psi\rangle = \otimes_{i=1}^{n} g_i |\psi\rangle$ can be mapped to another state in the same SLOCC class, $h |\psi\rangle = \otimes_{i=1}^{n} h_i |\psi\rangle$, via SEP if and only if there exists a set of probabilities $\{p_i\}_{i=1}^{m}$ such that

$$\frac{1}{r} \sum_{i=1}^{m} p_i S_i^\dagger H S_i + g^\dagger \sum_j N_j^\dagger N_j g = G, \tag{3}$$

where $S_i \in \mathcal{S}_\psi$, $N_j \in \mathcal{N}_{g\psi}$, $H = h^\dagger h$, $G = g^\dagger g$ and $r = ||h |\psi\rangle||^2 / ||g |\psi\rangle||^2$. We will use this notation of $G$ and $H$ throughout this paper, with $g$ and $G$ always referring to the initial state, and $h$ and $H$ always referring to the final state of the transformation. The invertible Kraus operators of this SEP map are given by $M_i = (\sqrt{p_i/r}) h S_i g^{-1}$, and thus each Kraus operator can be identified with a unique element of the stabilizer.

If a SEP transformation is possible without the use of operators which annihilate the initial state, then the transformation is said to be possible via $\text{SEP}_1$. It must then satisfy

$$\frac{1}{r}\sum_{i=1}^{m} p_i S_i^\dagger H S_i = G \,. \tag{4}$$

It was shown in Ref. [29] that, considering pure state transformations, $\text{LOCC}_\mathbb{N}$ protocols (those which terminate after a finite number of rounds) are a strict subset of $\text{SEP}_1$. However, it remains an open question whether LOCC is a subset of $\text{SEP}_1$.

As we already pointed out, Eq. (3) highlights the importance of local symmetries for SEP (and therefore LOCC) transformations. In particular, in [28], it was shown that if the stabilizer of a state $|\psi\rangle$ is trivial (i.e. $\mathcal{S}_\psi = \{\mathbb{1}\}$), then no LOCC transformation from this state to any other pure state is possible. Moreover, all states in the SLOCC class of $|\psi\rangle$ are then isolated under LOCC, in the sense that they can neither be reached from, nor be transformed into, any other state.

This has considerable implications for entanglement theory, as it was shown in [28, 30] that in homogeneous systems of at least four parties (five parties for qubit systems), states are generically in an SLOCC class with trivial stabilizer. Consequently, in these systems, almost all states are isolated under LOCC and the MES is therefore of full measure.

In light of these results, LOCC transformations between multipartite states appear to be rather exceptional. For this reason, one might want to relax some of the constraints and consider, for example, probabilistic transformations. This has been done for both SEP and LOCC transformations in the literature [53]. We highlight here some important results about the maximum success probability of such transformations. Considering a probabilistic SEP transformation from a state $g|\psi\rangle$ to another state $h|\psi\rangle$, and using the same notations as before, the maximum probability of success for this transformation, $p_{\max}^{\text{SEP}}$, is given by[1] [53]

$$p_{\max}^{\text{SEP}} = \max\left\{ \sum_i p_i : rG - \sum_i p_i S_i^\dagger H S_i \in \text{sep} \right\}, \tag{5}$$

where sep denotes the set of separable operators.

Adding some additional assumptions can make this success probability easier to compute. For instance, if the stabilizer of $g|\psi\rangle$ is finite and unitary, then the previous equation reduces to

$$p_{\max}^{\text{SEP}} = \max\left\{ p : rG - \frac{p}{|\mathcal{S}_\psi|}\sum_i S_i^\dagger H S_i \in \text{sep} \right\}, \tag{6}$$

where $|\mathcal{S}_\psi|$ is the number of elements in the stabilizer of $|\psi\rangle$.

Alternatively, if the stabilizer of a normalized seed state is unitary, then the maximum success probability of reaching the seed state (i.e. $H = \mathbb{1}$) from any state $g|\psi\rangle$ is given by [53]:

$$p_{\max}^{\text{SEP}}(g|\psi\rangle \mapsto |\psi\rangle) = \lambda_{\min}\left[ \frac{G}{||g|\psi\rangle||^2} \right], \tag{7}$$

where $\lambda_{\min}[M]$ corresponds to the minimum eigenvalue of the operator $M$.

It was further shown in Ref. [28] that, if the stabilizer of $|\psi\rangle$ is trivial (as is particularly relevant, as states in homogeneous systems generically have trivial stabilizer), then this transformation can be implemented by an LOCC one-successful-branch protocol (OSBP) with probability equal to $p_{\max}^{\text{SEP}}$. Thus, for these transformations, we have $p_{\max}^{\text{SEP}} = p_{\max}^{\text{LOCC}}$. Note further

---

[1]Note that although they do not appear in the formula for computing $p_{\max}^{\text{SEP}}$, this result also holds when taking into account operators from $\mathcal{N}_{g\psi}$, i.e. operators that annihilate the initial state. This is because, for probabilistic SEP transformations, these singular operators can be included in a branch of the transformation that fails.

that, thus, Eq. (7) fully characterizes the maximal success probability of any transformation within an SLOCC class with trivial stabilizer.

This concludes our review of the tools that will be used throughout this paper. In the next section, we give further details about the multi-state extension of LOCC that we investigate in this paper.

## 3 Setting the stage and General observations

As mentioned in the introduction, the main goal of this paper is to investigate LOCC transformations on multiple states as an extension of LOCC. We refer to this as the multi-state regime. In this regime, given a target state $|\psi\rangle$, from some Hilbert space $\mathcal{H}$, we want to investigate which state $|\phi\rangle$ the state $|\psi\rangle$ can be transformed into if we append to it, as an additional resource, an auxiliary state $|\bar{\psi}\rangle$ and perform joint LOCC on the two states (see Fig. 1).

In a multi-state transformation, each party is still constrained to act locally, i.e. only on the particles they control, and can classically communicate their measurement outcomes to the other parties. Such a regime is physically motivated: if the $n$ parties have access to a quantum memory, they may store resourceful auxiliary states and then use them to transform the target state. We impose only that the state-splitting is preserved after the transformation, i.e. that the transformation takes the form

$$|\psi\rangle \otimes |\bar{\psi}\rangle \rightarrow |\phi\rangle \otimes |\bar{\phi}\rangle \,, \tag{8}$$

for some state $|\bar{\phi}\rangle$. If such a transformation is possible, we say that $|\psi\rangle$ is LOCC transformable to $|\phi\rangle$ with the help of an auxiliary system. As our intent is to better understand the multipartite entanglement contained in $|\psi\rangle$ and/or to relate it to the one contained in $|\phi\rangle$, we consider both states to belong to the same Hilbert space[2]. Note that specific types of multi-state transformations may be used to induce a new partial order. Specifically, if, for some $|\psi\rangle$ that cannot be transformed to $|\phi\rangle$, $k$ copies of $|\psi\rangle$ can be transformed to $k$ copies of $|\phi\rangle$, then considering LOCC under $k$ copies will induce a different partial order in the Hilbert space than single-state LOCC. Alternatively, if we constrain the auxiliary state to remain invariant, then the transformation corresponds to entanglement catalysis (which has been extensively studied for bipartite states [43,54]). Let us emphasize at this point that, contrary to other approaches (see for instance Refs. [34,40]), we are working in the deterministic, non-asymptotic regime.

To achieve a multi-state transformation, we could in principle use auxiliary states $|\bar{\psi}\rangle$ and $|\bar{\phi}\rangle$ from an arbitrary Hilbert space, $\bar{\mathcal{H}}$, not necessarily identical to $\mathcal{H}$. As mentioned before, the new transformations that become possible in this regime (compared to the single-state regime) would then naturally depend on the dimension of this Hilbert space, $\bar{\mathcal{H}}$. If $\dim(\bar{\mathcal{H}}) > \dim(\mathcal{H})$, we have to take into account LOCC protocols transforming an auxiliary state $|\bar{\psi}\rangle$ from the higher dimensional Hilbert space $\bar{\mathcal{H}}$ to a state $|\phi\rangle$ of the smaller dimensional Hilbert space $\mathcal{H}$, as such transformations would straightforwardly lead to a multi-state transformation from $|\psi\rangle$ to $|\phi\rangle$. However, considering such transformations constitutes a different problem, that was put forward in Ref. [35]. As in the single-copy LOCC regime, our aim is to sort the entanglement contained in states belonging to the same Hilbert space, we do not want to focus on such transformations here. However, we will make use of interesting results from that setting later. For this reason, we choose to consider auxiliary states belonging to the same Hilbert space as the target state[3], or to a Hilbert space $\bar{\mathcal{H}}$ that corresponds to finitely many copies of the target state. This is physically motivated by the fact that, if one can store one auxiliary state

---

[2]This is also why we focus on the setting where the final target and auxiliary state factorize.

[3]We could also consider an auxiliary state from a Hilbert space $\bar{\mathcal{H}}$ with $\dim(\bar{\mathcal{H}}) < \dim(\mathcal{H})$, but this would exclude considering target states of qubits, which we want to avoid here.

to perform a multi-state transformation, one may also be able to store finitely many auxiliary states.

In this setting, if a state, $|\psi\rangle$, can be transformed into a state, $|\phi\rangle$, using $k-1$ auxiliary states from the same Hilbert space, we say that $|\psi\rangle$ can be transformed into $|\phi\rangle$ via $k$-LOCC. In any $k$-LOCC protocol, we can always consider a transformation that swaps the target state with one of the auxiliary states. Such a transformation does not combine the resources of the target and auxiliary states to achieve a new transformation but merely replaces the target state by a possibly more resourceful auxiliary state. For this reason, we consider such transformations as trivial and ignore them in the following. Let us also note here that, for sufficiently large $k$, $k$–LOCC transformations become trivial, as the auxiliary states can be used to distill Bell pairs between the parties, which can then be utilized to generate an arbitrary final target state via teleportation. However, such transformations are completely independent of the initial target state (implying that we cannot gain any additional knowledge about the entanglement contained in the state), utilize only bipartite entanglement and consume many resources. Thus, we also disregard them in the following work.

We expect that multi-state LOCC provides new non-trivial transformations. That is, for some impossible transformations, $|\psi\rangle \nrightarrow_{LOCC} |\phi\rangle$, we expect that there exists a state, $|\bar{\psi}\rangle$ (that cannot be transformed into $|\phi\rangle$ by LOCC), which enables the transformation in Eq. (8). One question which immediately reveals itself is whether these new transformations reduce the set of states required to reach all states in the Hilbert space, i.e. whether it makes the MES smaller. Before discussing other relevant questions in this context, let us address this one first: Is it possible that any reasonable generalisation of the MES to $k$-LOCC (i.e. a set of states that reaches all states and which is in some sense minimal) is different from the original MES.

The bipartite setting already reveals a feature of such a generalized MES; namely, that it may not be unique. Indeed, for bipartite states, it is well-known that the maximally entangled state can be reached via LOCC from two identical copies of a non-maximally entangled state (provided that this state is sufficiently entangled). Therefore, any bipartite state (of the same Hilbert space) can be reached from these two copies, and any set consisting only of one such non-maximally entangled state could be a 2-MES for this system. With this in mind, we define a multi-state MES as follows. A set $\mathcal{M}_k$ (not necessarily finite) containing states from a Hilbert space $\mathcal{H}$, is a $k$-MES if (a) any state of $\mathcal{H}$ can be reached via $k$-LOCC from $k$ (not necessarily distinct) states chosen in $\mathcal{M}_k$; and (b) it is minimal, in the sense that no strict subset of $\mathcal{M}_k$ satisfies (a).

Naturally, the 1-MES corresponds to the original MES from Ref. [26]. In that work, it was noted that the MES can equivalently be defined as the set containing all states that are not reachable via LOCC (from a different initial state) in a given Hilbert space. However, this equivalence of definitions is valid only for single-state transformations, and the alternative definition cannot be used for $k > 1$. Indeed, as our previous discussion of bipartite multi-state transformations indicates, the maximally entangled state (and thus any bipartite state) can be reached via a non-trivial multi-state LOCC transformation. This implies that the alternative definition for the $k$-MES of bipartite systems would lead to an empty set for all $k \geq 2$.

We also note here that a $k$-MES can always be chosen as a subset of the 1-MES. Indeed, the 1-MES allows one to reach all states via 1-LOCC, and therefore also via $k$-LOCC. We then obtain a $k$-MES by finding a minimal subset of the 1-MES preserving this property for $k$-LOCC. However, depending on the situation, one might prefer to chose a $k$-MES consisting of less-entangled states outside the MES, as they may be easier to produce experimentally. Finally, taking the discussion on distilling Bell pairs one step further, note that for sufficiently large $k$, the $k$-MES may always be chosen as a set containing only a single state.

In the general framework of multi-state transformations, several important questions arise. Is it possible to non-trivially change the SLOCC class of the target state? Are there unexpected

new transformations, such as transformations allowing one to reach a state from the MES from states that do not belong to the MES? Does $k$-LOCC always provide new transformations of a given target state? Does the multi-state regime also improve probabilistic transformations?

We answer the first question in Section 4.1 by showing that, in fact, the SLOCC class of the target state can be non-trivially changed via a multi-state LU transformation. This result implies that multi-state transformations cannot be fully characterized by using only the tools that have been developed to study single-state LOCC transformations. It also reveals that the multi-state regime provides a much richer set of new transformations. In Section 4.2, we answer the second question by showing that the 3-qubit GHZ state (which is in the MES) can be reached from two states that are not in the 3-qubit MES. In this section, we also show that certain combinations of target and auxiliary states can only achieve trivial transformations, which hints towards a negative answer to the third question. Finally, we provide a positive answer to the fourth question, by showing that the maximum success probability of a multi-state transformation transforming two states simultaneously can be greater than the maximum success probability of transforming these two states independently; in fact, we show it can even be greater than the maximum success probability of either single-state transformation.

In the following, special emphasis is given to transformations in which the final auxiliary states are fully-entangled. Such transformations exclude the trivial possibility of using teleportation, and have the advantage of not being wasteful with entanglement, in the sense that all states remain fully-entangled after the transformation. We study such transformations in the next section, where we investigate how the additional resource of 2-LOCC affects LOCC transformations of multipartite target states.

## 4 Multi-state Multipartite LOCC

We study here, for multipartite states, the problem of 2-LOCC transformations, posed in Eq. (8). Throughout the following sections, we highlight several features of multi-state LOCC showing that, already in the two-state regime, multi-state LOCC offers a much richer landscape of transformations than single-state LOCC. We start, in the next section, by discussing how multi-state LOCC allows one to non-trivially change the SLOCC class of the target state. We naturally exclude the trivial possibility of changing the SLOCC class of a state through a projective measurement, as such a transformation would merely destroy some of the entanglement of the state, and not genuinely change its type of entanglement.

### 4.1 Changing SLOCC class

We show here that multi-state LOCC allows one to deterministically change the SLOCC class of the target state with only LUs. As stated below, we show in addition that it is possible to change the orbit type of the SLOCC class of the target state.

**Observation 1.** *It is possible to change the SLOCC class of the initial states under multi-state LU transformations. Furthermore, the orbit type of the SLOCC class of these states can also be changed.*

This can be seen by considering a transformation of the form of Eq. (8), with

$$|\psi\rangle = |\text{GHZ}\rangle^{\otimes 2} , \tag{9}$$

$$|\bar{\psi}\rangle = |\text{W}\rangle^{\otimes 2} , \tag{10}$$

$$|\phi\rangle = |\bar{\phi}\rangle = |\text{GHZ}\rangle |\text{W}\rangle . \tag{11}$$

All these states should be understood as three-partite states which are shared among the parties as shown in Fig. 2. In this transformation, the target state has changed SLOCC class. This can be seen by observing the target state has in fact changed SLOCC orbit type: $|GHZ\rangle^{\otimes 2}$ is a critical state, whereas $|GHZ\rangle\,|W\rangle$ belongs to the null-cone (see the preliminaries). The proof of the observation follows then by noticing that such a transformation can be achieved with LU operations that permute the basis states of all parties in such a way that some of the local dimensions are swapped between the target and auxiliary states (see Fig. 2). We refer to such a transformation as a "sub-SWAP" transformation. In Appendix A, we present more details on how the orbit type of SLOCC classes may change under multi-state LU transformations.

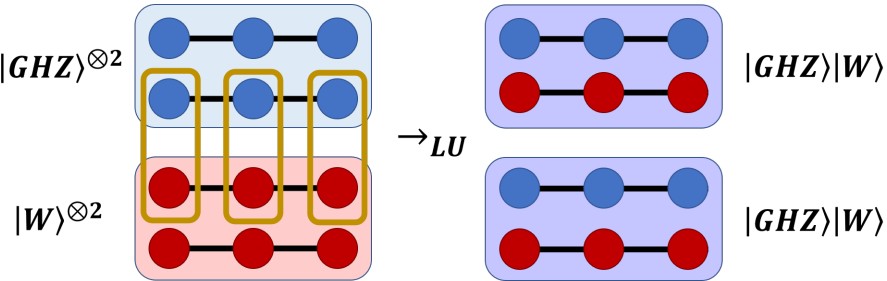

Figure 2: Example of a multi-state transformation changing the SLOCC class of the initial states using LU operations. The initial states both consist of a tensor product of two states. By applying LUs (indicated by beige boxes), one can SWAP one of the GHZ states with one of the W states, yielding two copies of a new state that is in a different SLOCC class to both initial states.

Clearly, more involved instances transforming states that are genuinely entangled across all local dimensions can be easily generated from this example by adding LUs acting on the target and auxiliary states separately, before and after the transformation given here.

This first observation confirms that LOCC in the multi-state regime is more complex than in the single-state regime, and one cannot simply transfer over the methods from the single-state case. In the single-state case, transformations within an SLOCC class can be parameterized by using a single seed state and local invertible operators. As this result shows, in the multi-state case we generically have to consider an infinite number of possible seed states representing the possible SLOCC classes of the final states. Thus, there is no natural way to implement the condition that the final target and final auxiliary states factorize. For these reasons, a complete characterization of multi-state LOCC transformations seems very challenging.

Despite all that, it would be interesting to know whether this setting also enables new transformations of the target state within its SLOCC class. In particular, it is important to investigate whether states from the MES can be reached in the multi-state regime. We answer this question in the next section.

### 4.2 Transformations within the same SLOCC class

In this section, we consider multi-state transformations in which the individual states all belong to the same SLOCC class. In this case, it is handy to relate all the states to the same seed state $|\psi\rangle$, through local invertible operators. In this framework, we look for transformations of the form

$$g_1\,|\psi\rangle \nrightarrow_{LOCC} h_1\,|\psi\rangle$$
$$g_1\,|\psi\rangle \otimes g_2\,|\psi\rangle \rightarrow_{LOCC} h_1\,|\psi\rangle \otimes h_2\,|\psi\rangle \;, \tag{12}$$

where $g_{1,2}$ and $h_{1,2}$ are local invertible operators.

As mentioned in the preliminaries section, local symmetries play an essential role in LOCC transformations of multipartite states. States without non-trivial local symmetries are isolated under SEP and LOCC, which is a generic property among multipartite states [28,30]. However, in the multi-state case, even if the target and auxiliary states have a trivial stabilizer, their tensor product could have non-trivial symmetries, which could lead to a non-trivial multi-state transformation. For the case we consider in this section, in which the target and auxiliary states are in the same SLOCC class, the seed state $|\psi\rangle \otimes |\psi\rangle$ has always at least one additional local symmetry: $\text{SWAP}^{\otimes n}$ (which corresponds to permuting the target and auxiliary seed states, and which we refer to in the following simply as SWAP). In the following theorem, we show that this additional symmetry alone is not enough to provide new $\text{LOCC}_{\mathbb{N}}$ transformations.

**Theorem 2.** *Given a fully-entangled state $|\psi\rangle$ such that*

$$\mathcal{S}_{\psi^{\otimes 2}} = \{\mathbb{1}, \text{SWAP}\}, \tag{13}$$

*all transformations of the form*

$$g_1 |\psi\rangle \otimes g_2 |\psi\rangle \rightarrow_{LOCC_{\mathbb{N}}} h_1 |\psi\rangle \otimes h_2 |\psi\rangle \tag{14}$$

*are necessarily trivial. Moreover, if the final states are identical, i.e. $h_1 = h_2$, then the statement also holds for LOCC.*

*Proof.* First, we show that via $\text{LOCC}_{\mathbb{N}}$, there are only trivial transformations. As discussed in the preliminaries, if a transformation is possible via $\text{LOCC}_{\mathbb{N}}$, it is possible via $\text{SEP}_1$ [29]. Therefore, by Eq. (4) we have:

$$pH_1 \otimes H_2 + (1-p)H_2 \otimes H_1 = G_1 \otimes G_2 . \tag{15}$$

As $G_{1,2}$ are strictly positive operators, we have $\text{tr } G_{1,2} \neq 0$. Therefore, by taking the partial traces of Eq. (15), we can express $G_1$ and $G_2$ in terms of $p, H_1, H_2$. Re-inserting this into Eq. (15) yields either $G_1 \propto H_1$ and $G_2 \propto H_2$, or $G_1 \propto H_2$ and $G_2 \propto H_1$. Thus, the transformation is trivial.

Second, we show there are no non-trivial LOCC transformations if the final states are identical, i.e. if $h_1 = h_2$. If the transformation is possible via LOCC, it is possible via SEP. Therefore, we consider Eq. (3) (with $r = 1$, as the elements of the stabilizer are unitary[4]). Acting with both sides of this equation on $|\psi\rangle^{\otimes 2}$ yields:

$$(G_1^{-1} \otimes G_2^{-1})(H_1 \otimes H_1)|\psi\rangle^{\otimes 2} = |\psi\rangle^{\otimes 2} . \tag{16}$$

Therefore, $(G_1^{-1}H_1 \otimes G_2^{-1}H_1)$ is a local invertible symmetry of $|\psi\rangle$ and thus must belong to the stabilizer $\mathcal{S}_{\psi^{\otimes 2}}$. Moreover, it is separable in the state splitting. Therefore it must be equal to $\mathbb{1}$. Thus, it has to hold that, $G_1 \propto G_2 \propto H_1$. Hence, the transformation is trivial. $\qquad \square$

This theorem indicates that, given a state $|\psi\rangle$, if the stabilizer of $|\psi\rangle^{\otimes 2}$ consists of only $\mathbb{1}$ and SWAP, then only trivial transformations are possible. Consequently, we refer to such stabilizers (i.e. those satisfying Eq. (13)) as trivial. We will give an explicit example of a state with a trivial stabilizer in Section 4.4. To find non-trivial transformations within a single SLOCC class in the multi-state regime, one should consider SLOCC classes represented by a seed state $|\psi\rangle$, such that $|\psi\rangle^{\otimes 2}$ has a non-trivial stabilizer. As we show now, SLOCC classes of generalized GHZ states satisfy precisely this requirement, making them good candidates to study multi-state transformations.

---

[4]When all the elements of the stabilizer are unitary, Eq. (3) implies $r = \text{tr}(H)/\text{tr}(G)$. Without loss of generality, we can thus choose to normalize the operators $H$ and $G$ so that $r = 1$.

A generalized GHZ state for $n$ parties with local dimension $d$, that we denote by $|\text{GHZ}_d^n\rangle$, corresponds to the state

$$|\text{GHZ}_d^n\rangle = \frac{1}{\sqrt{d}} \sum_{i=0}^{d-1} |\underbrace{i \cdots i}_{n}\rangle \,. \tag{17}$$

Such states have the useful property that $k$ copies of them can be re-expressed as another generalized GHZ state with the same number of parties but higher local dimensions. The $k$ copies $|\text{GHZ}_d^n\rangle^{\otimes k}$ are indeed equivalent, up to a local relabeling of the computational basis states, to the state $|\text{GHZ}_{d^k}^n\rangle$.

As a consequence, computing the stabilizer of a generalized GHZ state for any local dimension is sufficient to obtain the stabilizer of any number of copies of a generalized GHZ state. This stabilizer can easily be computed and is given in the following lemma (see Appendix B for the proof).

**Lemma 3.** *A local invertible operator is a symmetry of the state* $|\text{GHZ}_d^n\rangle$ *(with $d \geq 2$ and $n \geq 3$) if and only if it can be written*

$$S = \left[ D(\vec{\gamma}^{(1)}) \otimes \cdots \otimes D(\vec{\gamma}^{(n)}) \right] X_\sigma^{\otimes n} \,, \tag{18}$$

*where*

$$D(\vec{\gamma}^{(i)}) = \text{diag}(\gamma_1^{(i)}, \gamma_2^{(i)}, \ldots, \gamma_d^{(i)}) \,, \tag{19}$$

$$\gamma_j^{(n)} = \left( \prod_{i=1}^{n-1} \gamma_j^{(i)} \right)^{-1} \,, \quad \forall j \in \{1, \ldots, d\} \tag{20}$$

$$X_\sigma = \sum_{k=0}^{d-1} |\sigma(k)\rangle \langle k| \,, \tag{21}$$

*with $\sigma \in S_d$ any permutation of $d$ elements, $\vec{\gamma}^{(i)} = (\gamma_1^{(i)}, \ldots, \gamma_d^{(i)}) \in \mathbb{C}^d$ for $i = 1, \ldots, n-1$.*

Knowing the symmetries of all states $|\text{GHZ}_d^n\rangle$, we can fully characterize single-state LOCC transformations among states in a subset of their SLOCC classes. We will later use this to prove some interesting properties of the multi-state regime. In particular, we show in Theorem 5, that LOCC transformations between states of the form

$$\mathbb{1} \otimes \cdots \otimes \mathbb{1} \otimes g \, |\text{GHZ}_d^n\rangle \,, \tag{22}$$

with $g = \text{diag}(g_1, \ldots, g_d)$ any invertible diagonal matrix, obey a majorization condition (just like for bipartite states). Recall, a real vector $a = (a_1, \ldots, a_m)^T$ majorizes another real vector $b = (b_1, \ldots, b_m)^T$, denoted as $a \succ b$, if

$$\sum_{i=1}^{k} a_i^\downarrow \geq \sum_{i=1}^{k} b_i^\downarrow, \quad \forall \, k \in \{1, \ldots, m\} \,, \tag{23}$$

with equality in the case of $k = m$, and where $a^\downarrow = (a_1^\downarrow, \ldots, a_m^\downarrow)^T$, $b^\downarrow = (b_1^\downarrow, \ldots, b_m^\downarrow)^T$ correspond to the vectors $a$ and $b$ resorted into descending order. We now present a matrix reformulation of a theorem by Rado [55]:

**Theorem 4** ( [55]). *Given two real diagonal matrices $A = \text{diag}(a_1, \ldots, a_d)$ and $B = \text{diag}(b_1, \ldots, b_d)$ of dimension $d$, there exists a probability distribution, $\{p_k\}_{k=1}^m$, such that*

$$\sum_{k=1}^{m} p_k X_{\sigma_k} A X_{\sigma_k}^\dagger = B \,, \tag{24}$$

*where each index $k$ represents a permutation $\sigma_k \in S_d$, with associated permutation operator, $X_{\sigma_k}$, if and only if*

$$(a_1, ..., a_d)^T \succ (b_1, ..., b_d)^T \, . \tag{25}$$

This theorem is the central tool for proving the following result:

**Theorem 5.** *Let $g = diag(g_1, \ldots, g_d)$ and $h = diag(h_1, \ldots, h_d)$ be two invertible, complex, diagonal matrices such that $\mathrm{tr}(g^\dagger g) = \mathrm{tr}(h^\dagger h)$. Then the transformation*

$$\mathbb{1} \otimes \cdots \otimes \mathbb{1} \otimes g \, |\mathrm{GHZ}_d^n\rangle \xrightarrow{LOCC} \mathbb{1} \otimes \cdots \otimes \mathbb{1} \otimes h \, |\mathrm{GHZ}_d^n\rangle \, , \tag{26}$$

*exists if and only if*

$$(|h_1|^2, \ldots, |h_d|^2)^T \succ (|g_1|^2, \ldots, |g_d|^2)^T \, .$$

*Proof.* (*only if*) If the transformation is possible via LOCC, it is necessarily also possible via SEP. Therefore, we may consider the necessary and sufficient conditions for the existence of a SEP transformation, given in Eq. (3). In this case, because $\mathrm{tr}(g^\dagger g) = \mathrm{tr}(h^\dagger h)$ and the GHZ state is normalised, it is easy to see that we must have $r = 1$. From this operator equation, let us consider the sum of matrix elements $\sum_{i=0}^{d-1} \langle l, \ldots, l | \cdot | i, \ldots, i \rangle$, for some $l \in \{0, \ldots, d-1\}$. On the LHS, the second sum vanishes because any operator $N_j g$ annihilates the state $\sum_{i=0}^{d-1} |i, \ldots, i\rangle \propto |\mathrm{GHZ}_d^n\rangle$, and we get

$$\sum_{k=1}^m p_k \sum_{i=0}^{d-1} \langle l, \ldots, l | S_k^\dagger (\mathbb{1} \otimes \cdots \otimes \mathbb{1} \otimes H) S_k | i, \ldots, i \rangle \, . \tag{27}$$

Evaluating this for the symmetries as given in Eq. (18) yields:

$$\sum_{k=1}^m p_k \langle l | X_{\sigma_k}^\dagger H X_{\sigma_k} | l \rangle \, . \tag{28}$$

From this equation, we see that this sum of matrix elements is independent of the diagonal part of the symmetries, $D(\vec{\gamma}^{(i)})$. For the RHS, as $G$ is a diagonal matrix, the same sum of matrix elements merely reads $\langle l | G | l \rangle$. Since these equations are valid for all $l = 1, \ldots, d$, and $H$ and $G$ are diagonal, combining the left- and right-hand sides yields the matrix equation

$$\sum_{k=1}^m p_k X_{\sigma_k}^\dagger H X_{\sigma_k} = G \, , \tag{29}$$

which by Theorem 4 implies $(|h_1|^2, \ldots, |h_d|^2)^T \succ (|g_1|^2, \ldots, |g_d|^2)^T$.

(*if*) By Theorem 4, we know that there exist probabilities $\{p_k \geq 0\}_{k=1}^m$ with $\sum_{k=1}^m p_k = 1$ such that

$$\sum_{k=1}^m p_k X_{\sigma_k} H X_{\sigma_k}^\dagger = G \, . \tag{30}$$

Since the permutation operators $X_{\sigma_k}$, are symmetries of the seed state $|\mathrm{GHZ}_d^n\rangle$, this implies that the transformation can be done by $\mathrm{SEP}_1$ (see Eq. (4) in the preliminaries). To conclude the proof, we observe this transformation can also be achieved by an LOCC protocol in which the last party applies a measurement with measurement operators $\{\sqrt{p_k} h X_{\sigma_k} g^{-1}\}_{k=1}^m$ and then, depending on the outcome $k$, all the other parties apply the unitary operation $X_{\sigma_k}$.

$\square$

As already mentioned, generalized GHZ states have a structure that extends nicely to the multi-state regime. In fact, GHZ-like states of the form given in Eq. (22) admit a direct generalization of the bipartite Schmidt decomposition [56,57]. Using this generalized Schmidt decomposition, it was mentioned in Ref. [56] that the bipartite entanglement concentration protocol presented in [17] can straightforwardly be extended to concentrate the entanglement of GHZ-like states into perfect GHZ states (with an optimal asymptotic rate).

Using Theorem 5, we now show that, in the multi-state regime, it is possible to start with a target state and an auxiliary state that are outside the MES and use the auxiliary state to transform the target state into a state that is inside the MES[5].

To do so, we consider the following two-state transformation in the SLOCC class of two copies of the three-qubit GHZ state, $|\text{GHZ}\rangle \equiv |\text{GHZ}_2^3\rangle$ (note that the subsequent discussion can be immediately generalized to $n > 3$ parties):

$$|\tilde{g}\rangle \otimes |\tilde{g}\rangle \xrightarrow{LOCC} |\tilde{h}_1\rangle \otimes |\tilde{h}_2\rangle \,, \tag{31}$$

with

$$|\tilde{g}\rangle = (\mathbb{1} \otimes \mathbb{1} \otimes \tilde{g} )|\text{GHZ}\rangle \,, \tag{32}$$

$$|\tilde{h}_i\rangle = \left(\mathbb{1} \otimes \mathbb{1} \otimes \tilde{h}_i \right)|\text{GHZ}\rangle \ (i = 1, 2) \,, \tag{33}$$

where $\tilde{G} = \tilde{g}^\dagger \tilde{g} = \mathbb{1}/2 + \delta \sigma_z$ and $\tilde{H}_i = \tilde{h}_i^\dagger \tilde{h}_i = \mathbb{1}/2 + \alpha_i \sigma_z$ $(i = 1, 2)$, with $\delta, \alpha_1, \alpha_2 \in [0, 1/2)$. Observe that, by Theorem 5, the smaller the value of $\delta, \alpha_1$ or $\alpha_2$, the more entangled the corresponding state (see also [23]). In addition, it has been shown in Ref. [26] that, among states of this form, only the GHZ state is in the MES of three-qubit states.

In the LOCC transformation given in Eq. (31), the two copies of the GHZ state can equivalently be replaced by the state $|\text{GHZ}_4^3\rangle$, yielding a transformation of the same form as in Theorem 5, with the 4-dimensional invertible local matrices $g = \tilde{g} \otimes \tilde{g}$ and $h = \tilde{h}_1 \otimes \tilde{h}_2$. As, by construction, we have $\text{tr}(\tilde{g}^\dagger \tilde{g} \otimes \tilde{g}^\dagger \tilde{g}) = \text{tr}(\tilde{h}_1^\dagger \tilde{h}_1 \otimes \tilde{h}_2^\dagger \tilde{h}_2) = 1$, we can apply Theorem 5. It is easy to see that the corresponding majorization condition is satisfied if and only if

$$\delta \le \sqrt{\left(\alpha_1 + \frac{1}{2}\right)\left(\alpha_2 + \frac{1}{2}\right)} - \frac{1}{2} \,. \tag{34}$$

Note, that by Eq. (34), if we wish to reach two copies of a GHZ state (i.e. $\alpha_1 = \alpha_2 = 0$), then we must begin from two copies of a GHZ state ($\delta = 0$). Alternatively, setting $\alpha_1 = 0$ (which corresponds to transforming only the target state into the GHZ state) yields:

$$\delta|_{\alpha_1=0} \le \sqrt{\left(\frac{\alpha_2}{2} + \frac{1}{4}\right)} - \frac{1}{2} \le \alpha_2 \,. \tag{35}$$

This inequality has solutions $\forall \alpha_2 \in (0, 1/2)$. Therefore, provided Eq. (35) is satisfied by the initial states, we can transform the target state into the GHZ state. That is, in the multi-state regime, we can transform a state that is outside the MES to a state that is inside the MES.

Note that a simple teleportation-like protocol does not allow one to obtain $|\text{GHZ}\rangle$. That is a protocol in which the two initial states are first transformed into bipartite states by projectively measuring one particle. This may be easily seen by considering the entropies of the local reduced density matrices. Clearly, when considering more than three particles this holds all the more.

Note further that Eq. (35) tells us that, if we transform the target state to the GHZ state, then $\alpha_2 \ge \delta$. That is, after the transformation, the auxiliary state is less entangled (this is

---

[5]See [58] for a similar work investigating these types of transformations for the state $|W\rangle^{\otimes 2}$.

not surprising as the overall transformation is an LOCC transformation). Therefore, in the multi-state regime, it is possible to squeeze entanglement from one state to another.

As another consequence of this result, when considering the 2-MES of three-qubit states as in Section 3, a choice of the 2-MES that contains a state $|\tilde{g}\rangle$ as in Eq. (32), but not $|GHZ\rangle$ is thinkable. In fact, in Ref. [35] it was shown that $|GHZ_3^3\rangle$ may be transformed into any three-qubit state by LOCC. Hence, $\{|GHZ\rangle\}$ is a 2-MES for three-qubit states. Moreover, as two copies of $|\tilde{g}\rangle$ may be converted into $|GHZ_3^3\rangle$ (see Theorem 5 with slight modification allowing for non-invertible operators) as long as $\delta \leq 1/\sqrt{3} - 1/2 \approx 0.077$, the corresponding sets $\{|\tilde{g}\rangle\}$ also form a 2-MES for three-qubit states. This fact resembles the freedom in choosing the 2-MES in the bipartite case discussed in Section 3. Let us also remark here that—in contrast to the three-qubit system considered here—not for all system sizes it is possible to find a finite set forming a 2-MES. Indeed, a simple counting argument shows that for sufficiently large $n$, a finite set of states in $\left(\mathbb{C}^4\right)^{\otimes n}$ does not suffice to even probabilistically obtain all $n$-qubit states.

Recall that, as explained in the preliminaries, each Kraus operator in a SEP map is associated with a unique invertible symmetry from the stabilizer. Consequently, whether a transformation is possible under LOCC is intrinsically connected to the stabilizer of the state. Thus, one might ask: which are the relevant symmetries enabling a certain LOCC transformation? Novel transformations in the multi-state regime will naturally need local operations that act jointly on the two copies of the initial state, i.e. that are non-local in the state splitting. As discussed above, two copies of a state have at least one symmetry that is non-local in the state splitting: SWAP. As the stabilizer of $|\psi\rangle^{\otimes 2}$ always contains the symmetries that can be generated by SWAP and the single-copy symmetries of $|\psi\rangle$, we refer to these symmetries as trivial. These trivial symmetries in fact form a subgroup of the stabilizer, which we will refer to as the trivial subgroup $\mathcal{S}^0_{\psi^{\otimes 2}}$. Additionally, we refer to symmetries that cannot be generated by SWAP and the single-copy symmetries of $|\psi\rangle$ as emergent. As we will see in the following, the trivial subgroup does allow novel transformations in the multi-state regime. It is now natural to ask: is $\mathcal{S}^0_{\psi^{\otimes 2}}$ sufficient to implement all multi-state LOCC transformations?

We now show the answer to this question is no. That is, there are LOCC transformations which require emergent symmetries. To this end, we again consider transformations as in Eq. (31), but now only allowing measurement operators corresponding to elements of the trivial subgroup. Following the arguments in the proof of Theorem 5, Eq. (29) must hold for a transformation to be possible, where the sum is now over permutation matrices from the following subgroup of the trivial subgroup:

$$
\tilde{\mathcal{S}}^0_{GHZ^{\otimes 2}} = \{\mathbb{1},\, X \otimes \mathbb{1},\, \mathbb{1} \otimes X,\, X \otimes X,\, \text{SWAP}, \text{SWAP}.(X \otimes \mathbb{1}),
$$
$$
\text{SWAP}.(\mathbb{1} \otimes X),\, \text{SWAP}.(X \otimes X)\} \subseteq \mathcal{S}^0_{GHZ^{\otimes 2}},
\tag{36}
$$

which is a group of order 8, in contrast to the full permutation group of four elements, which is of order 24. Here, $X$ denotes the Pauli X and should be understood as $X^{\otimes 3}$, just as SWAP.

Evaluating Eq. (29) over this symmetry subgroup yields the following bound:

$$
\delta \leq \sqrt{\alpha_1, \alpha_2}.
\tag{37}
$$

Thus, we see that if we want to transform the target state to the GHZ state, i.e. $\alpha_1 = 0$, then $\delta$ must also be zero. That is, if we are restricted to trivial symmetries, we can only reach the GHZ state by starting with it[6].

As a final comment, note that all transformation saturating the inequality in Eq. (37) can be decomposed into a particularly simple two round protocol which only uses trivial symmetries.

---

[6]i.e. in order to reach $|GHZ\rangle$, we need non-local symmetries such as $X_{(13)} = |0\rangle\langle 0| + |3\rangle\langle 1| + |2\rangle\langle 2| + |1\rangle\langle 3|$.

Let $\lambda = \sqrt{\alpha_1 \alpha_2}/(\alpha_1 + \alpha_2) \in (0, 1/2)$. First, the last party applies a measurement with the following measurement operators:

$$M_1^1 = \sqrt{1/2}\, h' \left(g^{-1} \otimes g^{-1}\right) \tag{38}$$

$$M_2^1 = \sqrt{1/2}\, h'\, \mathrm{SWAP}\left(g^{-1} \otimes g^{-1}\right), \tag{39}$$

where $h' = \sqrt{H'}$ with:

$$H' = \left(\frac{1}{2} + \lambda\right)\tilde{H}_1 \otimes \tilde{H}_2 + \left(\frac{1}{2} - \lambda\right) X\tilde{H}_1 X \otimes X\tilde{H}_2 X. \tag{40}$$

Using Eq. (29), it is easy to verify that $\{M_i^1\}_{i=1}^2$ forms a valid measurement. In the event of outcome 1, the parties do nothing, and, in the event of outcome 2, parties 1 to 2 apply a SWAP. Thus, this first round of the LOCC protocol deterministically transforms the initial state, $\mathbb{1} \otimes \mathbb{1} \otimes (g \otimes g)|\mathrm{GHZ}_4^3\rangle$, into the state $\mathbb{1} \otimes \mathbb{1} \otimes h'|\mathrm{GHZ}_4^3\rangle$ (which, we might note, is not state separable). Next, the last party applies a second measurement with measurement operators:

$$M_1^2 = \sqrt{1/2 + \lambda}\,\left(\tilde{h}_1 \otimes \tilde{h}_2\right) h'^{-1} \tag{41}$$

$$M_2^2 = \sqrt{1/2 - \lambda}\,\left(\tilde{h}_1 \otimes \tilde{h}_2\right)(X \otimes X)\, h'^{-1}, \tag{42}$$

which by construction satisfies the completeness relation. In the event of outcome 1, the parties do nothing, and, in the event of outcome 2, parties 1 and 2 apply $X \otimes X$. Thus, the second round deterministically transforms the state $\mathbb{1} \otimes \mathbb{1} \otimes h'|\mathrm{GHZ}_4^3\rangle$ to the final state, $\mathbb{1} \otimes \mathbb{1} \otimes (\tilde{h}_1 \otimes \tilde{h}_2)|\mathrm{GHZ}_4^3\rangle$. Observe, all measurements throughout the protocol only depend on trivial symmetries from the subgroup, $\tilde{\mathcal{S}}_{\mathrm{GHZ}^{\otimes 2}}^0$. Moreover, although the symmetries used are separable, each measurement is non-local in the state splitting.

## 4.3 Multipartite LOCC Catalysis

Theorem 5 shows that for a class of GHZ-like states, LOCC transformations are fully characterized by a majorization condition, just like they are for bipartite states. We can therefore use this fact to provide, to our knowledge, the first examples of multipartite catalytic transformation. In the following we present an explicit example. Let us consider two 4-dimensional GHZ-like states over $n$ parties $|\psi_1\rangle$ and $|\psi_2\rangle$, characterized by the matrices $g = \mathrm{diag}(\sqrt{0.45}, \sqrt{0.35}, \sqrt{0.12}, \sqrt{0.08})$ and $h = \mathrm{diag}(\sqrt{0.56}, \sqrt{0.21}, \sqrt{0.17}, \sqrt{0.06})$, respectively, as in Theorem 4. Theorem 5 shows that $|\psi_1\rangle$ and $|\psi_2\rangle$ are LOCC incomparable. As catalyst, we consider another 4-dimensional GHZ-like state over $n$ parties $|\phi_c\rangle$, characterized by the diagonal matrix $c = \mathrm{diag}(\sqrt{0.63}, \sqrt{0.27}, \sqrt{0.07}, \sqrt{0.03})$. Using the fact that the tensor product state $|\mathrm{GHZ}_d^n\rangle \otimes |\mathrm{GHZ}_d^n\rangle$ is equivalent to the higher dimensional state $|\mathrm{GHZ}_{d^2}^n\rangle$, we see that the catalytic transformation $|\psi_1\rangle \otimes |\phi_c\rangle \xrightarrow{LOCC} |\psi_2\rangle \otimes |\phi_c\rangle$ is equivalent to the transformation $\mathbb{1}_{16} \otimes \cdots \otimes \mathbb{1}_{16} \otimes (g \otimes c)|\mathrm{GHZ}_{16}^n\rangle \xrightarrow{LOCC} \mathbb{1}_{16} \otimes \cdots \otimes \mathbb{1}_{16} \otimes (h \otimes c)|\mathrm{GHZ}_{16}^n\rangle$. For the latter transformation, Theorem 5 applies and it is straightforward to verify that the corresponding majorization condition indeed holds.

Because of the relation to majorization, we can transfer another interesting result from bipartite state transformations to multipartite states. In Ref. [18], it has been shown that there exist bipartite states $|\psi\rangle$ and $|\phi\rangle$ such that neither $|\psi\rangle \to_{LOCC} |\phi\rangle$ nor $|\phi\rangle \to_{LOCC} |\psi\rangle$, yet $|\psi\rangle^{\otimes k} \to_{LOCC} |\phi\rangle^{\otimes k}$ for some $k \in \mathbb{N}$. As in the multipartite catalysis example above, choosing $\tilde{g}, \tilde{h}$ in Eqs. (32, 33) appropriately, we can reproduce these features in the multipartite case. Note, as discussed in the introduction, this implies the multi-state regime can induce a different partial order on the Hilbert space.

## 4.4 Multi-state probabilistic transformations

Finally, one might wonder whether the multi-state regime provides an advantage in probabilistic transformations. Such an advantage has been demonstrated for bipartite state transformations [18], where a simple expression for the maximal success probability of transformations is available [59]. However, the multipartite setting is naturally more complicated. Using results from Ref. [53] (see the preliminaries), we now demonstrate that the multi-state regime does indeed provide an advantage in probabilistically transforming two states together. Moreover, we demonstrate that the maximum success probability of the multi-state transformation can even be greater than the maximum success probability of either single-state transformation. Let us remark that, of course, the deterministic multi-state transformations presented above are already an example of this (with the success probability being 1 in the multi-state regime compared to strictly smaller than 1 otherwise). However, as Theorem 2 indicates, there may be states for which the multi-state regime allows no additional, non-trivial, deterministic transformations. We present an explicit example that demonstrates that, in such cases, probabilistic multi-state LOCC can still provide an advantage over probabilistic single-state LOCC.

Let $|\psi\rangle$ be a normalised state such that the stabilizer of $|\psi\rangle^{\otimes 2}$ only contains $\mathbb{1}$ and SWAP (note that, therefore, the stabilizer of $|\psi\rangle$ contains only $\mathbb{1}$), and let:

$$|\psi_i\rangle = \mathbb{1} \otimes \cdots \otimes \mathbb{1} \otimes h_i |\psi\rangle \, , \tag{43}$$

and $n_i = || |\psi_i\rangle ||$. Then by Eq. (6) [53] in the preliminaries we have:

$$p_{\max}^{SEP}\left(|\psi\rangle^{\otimes 2} \mapsto |\psi_1\rangle \otimes |\psi_2\rangle\right) \tag{44}$$

$$= \max\left\{p \text{ st } \mathbb{1}^{\otimes n-1} \otimes \left(\mathbb{1} - \frac{p}{2}\frac{H_1 \otimes H_2 + H_2 \otimes H_1}{(n_1 n_2)^2}\right) \text{ is sep}\right\} \tag{45}$$

$$= \lambda_{\max}^{-1}\left[\frac{H_1 \otimes H_2 + H_2 \otimes H_1}{2(n_1 n_2)^2}\right] . \tag{46}$$

Now consider the following choices for $H_i$:

$$H_1 = \begin{pmatrix} 1 & 0 \\ 0 & \epsilon \end{pmatrix}, \qquad H_2 = \begin{pmatrix} \epsilon & 0 \\ 0 & 1 \end{pmatrix} , \tag{47}$$

with $\epsilon \in (0, 1)$. Then we have:

$$\lambda_{\max}^{-1}\left[\frac{H_1 \otimes H_2 + H_2 \otimes H_1}{2(n_1 n_2)^2}\right] = \frac{2}{1 + \epsilon^2}(n_1 n_2)^2 > (n_1 n_2)^2 = \lambda_{\max}^{-1}\left[\frac{H_1}{n_1^2}\right]\lambda_{\max}^{-1}\left[\frac{H_2}{n_2^2}\right] . \tag{48}$$

Therefore, for all $\epsilon \in (0, 1)$, the maximum success probability of transforming both states at the same time is greater than the product of the maximum probabilities of each individual transformation by a factor of $2/(1+\epsilon^2)$. Note that the probability of the transformation depends on the norm of the final state, $n_i$, which in turn depends on $\epsilon$. We will discuss this further when we give a concrete example. First, we show that this maximum probability can be achieved via a multi-state probabilistic LOCC protocol. The LOCC protocol is given as follows: party $n$ performs a measurement with the following three measurement operators:

$$M_1 = \sqrt{\frac{1}{1+\epsilon^2}}\left(\mathbb{1}^{\otimes n-1} \otimes (h_1 \otimes h_2)\right) \tag{49}$$

$$M_2 = \sqrt{\frac{1}{1+\epsilon^2}}\left(\mathbb{1}^{\otimes n-1} \otimes (h_2 \otimes h_1)\right) \tag{50}$$

$$M_3 = \sqrt{\mathbb{1} - \left(M_1^\dagger M_1 + M_2^\dagger M_2\right)} , \tag{51}$$

where $M_3$ is positive semi-definite by virtue of Eq. (48). The last step of the protocol depends on the outcome of this measurement. If party $n$ gets outcome one, all parties do nothing; if they get outcome two, all parties apply a local SWAP; if they get outcome three, the protocol fails. As a consequence, the multi-state regime can provide an advantage in probabilistic LOCC transformations.

Taking a concrete example, we now illustrate how powerful the multi-state regime is in probabilistic transformations. Consider the state (from [30]):

$$|\psi\rangle = \frac{1}{\sqrt{22}}\Big(\sqrt{7}|00000\rangle + \sqrt{5}|11111\rangle + |00111\rangle + |01011\rangle + |01101\rangle + |01110\rangle + |10011\rangle$$
$$+ |10101\rangle + |10110\rangle + |11001\rangle + |11010\rangle + |11100\rangle\Big). \tag{52}$$

It can be verified that, in this case, $\mathcal{S}_{\psi^{\otimes 2}} = \{\mathbb{1}, SWAP\}$. Moreover, $n_i = \sqrt{(1+\epsilon)/2}$ (see Eq. (43)). Remarkably, this means that the probability of transforming both states simultaneously is given by:

$$p_{\max}\big(|\psi\rangle^{\otimes 2} \mapsto |\psi_1\rangle \otimes |\psi_2\rangle\big) = \frac{2}{1+\epsilon^2}\left(\frac{1+\epsilon}{2}\right)^2 > \frac{1+\epsilon}{2} = p_{\max}(|\psi\rangle \mapsto |\psi_i\rangle), \tag{53}$$

where the inequality is due to the fact $\epsilon \in (0,1)$.

That is, the probability of transforming both states simultaneously, is greater than the probability of even just one single-state transformation $\forall \epsilon \in (0,1)$. For example, the multi-state transformation has the greatest advantage over the single-state transformation if $\epsilon = 0.414$. In this case, the probability of transforming $|\psi\rangle \to_{LOCC} |\psi_1\rangle$ is 0.707 (therefore, the probability of independently transforming both $|\psi\rangle \to_{LOCC} |\psi_1\rangle$ and $|\psi\rangle \to_{LOCC} |\psi_2\rangle$ is 0.500). However, the probability of transforming both simultaneously, via a multi-state transformation, is 0.854.

Note that, unlike the probabilistic transformations in [28], such a transformation is not a One-Successful-Branch Protocol (OSBP). Moreover, in this example, $p_{\max}$ cannot in fact be achieved with an OSBP. This is because in an OSBP, by definition, the successful branch correspond to only one measurement operator, which in turn must correspond to an element of the stabilizer. As the stabilizer only contains $\mathbb{1}$ and SWAP, the $p_{\max}$ of this branch is at most the product of the maximum probabilities of each transformation.

Finally, note that, if $\epsilon = 0$, then $H_i$ become projectors, and thus $|\psi_i\rangle$ are no longer fully-entangled. Alternatively, $\epsilon = 1$, implies a trivial (deterministic) transformation and the multi-state regime provides no advantage (see Appendix C for further discussion for when the multi-state regime does not provide an advantage).

In summary, multi-state transformations can provide an advantage in probabilistically transforming two states together. In fact, the maximum success probability of transforming two states at the same time can be greater than transforming just one of them.

# 5 Bipartite multi-state LU Transformations

Bipartite entanglement, with its single SLOCC class of fully-entangled states and its unique maximally entangled state (up to LUs), has a very different structure compared to multipartite entanglement. As a result, some of the properties of the multi-state setting that we highlighted in the previous section do not apply for bipartite states. For instance, as we have fixed the dimension of the target state (thus avoiding trivial transformations such as $|\Phi_4^+\rangle|\Phi_4^+\rangle \to_{LU} |\Phi_8^+\rangle|\Phi_2^+\rangle$), it is not possible to change the SLOCC class of the target state. Since this possibility of changing SLOCC class is one of the main features that make the multi-state regime so difficult to characterize, bipartite systems seem to be a more reasonable setting

for trying to develop a systematic method to characterize the new transformations that emerge in the multi-state regime. As we saw in the previous section that even LU operations allow for a large set of new possible transformations, we consider in this section the simplest setting of two-state bipartite LU transformations. As we will see, even this very simple setting provides surprising results.

Throughout this section, we denote the target state by $|\mu\rangle \in \mathbb{C}^{d_\mu} \otimes \mathbb{C}^{d_\mu}$. As LU operations cannot transform a state to another state in a Hilbert space with a lower dimension, we consider in this section an auxiliary state, $|\lambda\rangle$, belonging to a Hilbert space, $\mathbb{C}^{d_\lambda} \otimes \mathbb{C}^{d_\lambda}$, of possibly larger dimension and ask whether there exist states $|\bar{\mu}\rangle \in \mathbb{C}^{d_\mu} \otimes \mathbb{C}^{d_\mu}$ and $|\bar{\lambda}\rangle \in \mathbb{C}^{d_\lambda} \otimes \mathbb{C}^{d_\lambda}$ such that

$$|\mu\rangle \nrightarrow_{LU} |\bar{\mu}\rangle \,,$$
$$|\mu\rangle \otimes |\lambda\rangle \rightarrow_{LU} |\bar{\mu}\rangle \otimes |\bar{\lambda}\rangle \,. \tag{54}$$

Note that a related problem has been studied in [60]. In that work, the aim was to find whether a high-dimensional bipartite or multipartite entangled state can be decomposed as a tensor product of lower-dimensional entangled states. Here, we start from a decomposable bipartite entangled state $|\mu\rangle \otimes |\lambda\rangle$ and search for all possible other decompositions of that state, in order to find non-trivial transformations of the target state that can be achieved through two-state LU. Because these transformations only involve bipartite states, it is more useful to describe them in terms of Schmidt coefficients. Let $\mu = (\mu_1, ..., \mu_{d_\mu})$ and $\lambda = (\lambda_1, ..., \lambda_{d_\lambda})$ denote the tuples of possibly degenerate, squared[7] Schmidt coefficients of the states $|\mu\rangle$ and $|\lambda\rangle$, respectively. Without loss of generality, we may sort all Schmidt coefficients in descending order and assume they are strictly positive, as zero-valued Schmidt coefficients can be removed by redefining the dimensions. Consequently, the bipartite state $|\mu\rangle \otimes |\lambda\rangle$ has strictly positive Schmidt coefficients given by the tuple $\mu \otimes \lambda = (\mu_1\lambda_1, ..., \mu_1\lambda_{d_\lambda}, \mu_2\lambda_1, ..., \mu_{d_\mu}\lambda_{d_\lambda})$. Similarly, the final state must also have a tensor product structure and can therefore be characterized by the tuple of Schmidt coefficients $\bar{\mu} \otimes \bar{\lambda}$, with $\bar{\mu} = (\bar{\mu}_1, ..., \bar{\mu}_{d_\mu})$ and $\bar{\lambda} = (\bar{\lambda}_1, ..., \bar{\lambda}_{d_\lambda})$ the Schmidt vectors of the final target and auxiliary states.

Applying any local unitary obviously cannot change the Schmidt coefficients of the state $|\mu\rangle \otimes |\lambda\rangle$; it can only change their order. As a consequence, an LU transformation from the state $|\mu\rangle \otimes |\lambda\rangle$ into the state $|\bar{\mu}\rangle \otimes |\bar{\lambda}\rangle$ corresponds to a non-trivial transformation of the target state $|\mu\rangle$ into an LU-inequivalent state $|\bar{\mu}\rangle$ if and only if there exist (ordered and normalized) sets of Schmidt coefficients $\bar{\mu} \neq \mu$, $\lambda$ and $\bar{\lambda}$ such that the $(d_\mu d_\lambda)$-tuple $\bar{\mu} \otimes \bar{\lambda}$ corresponds to a non-trivial permutation of the initial tuple $\mu \otimes \lambda$ (see Fig. 3). An upper bound for the number, $P$, of such permutations is given by (see e.g. [60]):

$$P \leq \frac{(d_\mu d_\lambda)!}{\prod_{i=1}^{d_\mu} \prod_{j=1}^{d_\lambda} (i+j-1)} \,. \tag{55}$$

To describe these transformations, we introduce the following equivalence relation: for any two $n$-tuples, $A$ and $B$, we say $A \sim B$ if the tuples are identical up to reordering. For example, $(1, 2, 2, 3) \sim (2, 3, 1, 2)$. With this notation, the transformation $|\mu\rangle \otimes |\lambda\rangle \rightarrow_{LU} |\bar{\mu}\rangle \otimes |\bar{\lambda}\rangle$ is possible if and only if

$$\mu \otimes \lambda \sim \bar{\mu} \otimes \bar{\lambda} \,. \tag{56}$$

From this observation, the problem we consider seems trivial, as it is simply equivalent to the problem of verifying the equivalence of two tuples. This problem is well-known and can, for example, be solved using the Elementary Symmetric Polynomials (ESPs) [61]. Generally

---

[7]Misusing notations for conciseness, we will refer to the squared Schmidt coefficients as Schmidt coefficients.

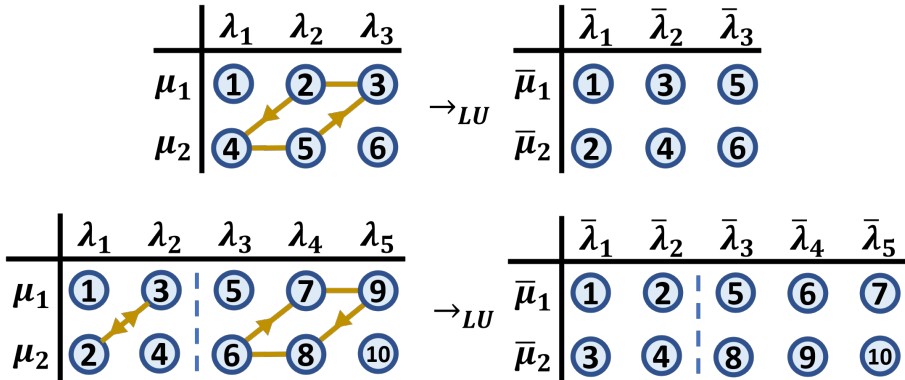

Figure 3: Non-trivial multi-state LU transformations correspond to non-trivial permutations of the Schmidt coefficients which preserve the tensor product structure. Here two non-trivial permutations are depicted. The circles with numbers represent the multiplied Schmidt coefficients (i.e. $\lambda_i \mu_j$) sorted into descending order. The first diagram corresponds to the transformation in Observation 7. The diagram below it corresponds to a "direct sum solution", as discussed after Theorem 6, that builds on this first transformation.

speaking, ESPs are indeed useful tools to study functions of several variables that do not depend on the order of these variables, see for instance Ref. [42]. Given a tuple of $n$ variables, $x = (x_1, ..., x_n)$, the elementary symmetric polynomial of degree $k$ over $x$, $e_k(x) \equiv e_k(x_1, ..., x_n)$, is defined as follows [61]:

$$e_k(x) \equiv \sum_{i_1 < i_2 < ... < i_k}^{n} x_{i_1} x_{i_2} .... x_{i_k}, \ \forall k = 1, ..., n. \tag{57}$$

In addition, we set $e_0 = 1$ and $e_k = 0, \ \forall k > n$. The ESPs provide simple necessary and sufficient conditions for two tuples to be identical up to reordering: for any two $n$-tuples $x$ and $y$, $x \sim y$ if and only if all their elementary symmetric polynomials are equal, i.e. if and only if $e_i(x) = e_i(y), \ \forall i \in \{1, ..., n\}$. As a consequence, bipartite two-state LU transformations can be studied in terms of ESPs over tuples of Schmidt coefficients. The necessary and sufficient condition of Eq. (56) for the transformation $|\mu\rangle \otimes |\lambda\rangle \to_{LU} |\bar{\mu}\rangle \otimes |\bar{\lambda}\rangle$ can equivalently be restated as

$$e_i(\mu \otimes \lambda) = e_i(\bar{\mu} \otimes \bar{\lambda}), \forall i \in \{1, ..., d_\mu d_\lambda\}. \tag{58}$$

These equations always admit the trivial solution $\bar{\mu} \sim \mu$ and $\bar{\lambda} \sim \lambda$ (corresponding to the identity permutation). Moreover, if $d_\mu = d_\lambda$, we have another trivial solution: $\bar{\lambda} \sim \mu$ and $\bar{\mu} \sim \lambda$ (corresponding to a SWAP of the states $|\mu\rangle$ and $|\lambda\rangle$). In the following, we again disregard these trivial solutions and only look for solutions leading to non-trivial transformations. The set of polynomial equations we have to solve grows quickly with the dimensions of the bipartite systems we consider, as it contains $d_\mu d_\lambda$ equations with even degrees ranging from 2 to $2d_\mu d_\lambda$. Determining all the solutions may therefore quickly become a difficult task. If we did not expect any non-trivial solutions for this set of equations, we could also use some powerful tools, such as the Positivstellensatz [62] from real algebraic geometry. This theorem indeed provides necessary and sufficient conditions for when a set of polynomial equalities, inequalities and inequations have no solutions. As we will show using a different approach, this method cannot directly be used here because there is an unexpectedly large set of different solutions.

We start, in the following section, by addressing this problem for the simplest case, in which the target state is restricted to a 2-qubit state. We fully characterize all the possible non-trivial

transformations of this target state. Building on this result, we show that a bipartite target state can always be non-trivially transformed using an auxiliary bipartite state of higher dimension. We then also use our characterization of qubit states transformations to show that when the auxiliary state has the same dimension as the target state non-trivial transformations can also always be achieved, except if the target and auxiliary states are 2-qubit or 2-qutrit states (in which case we prove no non-trivial transformations exists).

## 5.1 LU transformation in $(\mathbb{C}^2 \otimes \mathbb{C}^d)^{\otimes 2}$

In this section, we restrict the target state $|\mu\rangle$, to be a 2-qubit state and use an arbitrary 2-qudit auxiliary state. We use the notations presented in the previous section for the tuples of Schmidt coefficients and investigate non-trivial transformations of the target state under two-state LU. We thus only fix the 2-tuple $\mu = (\mu_1, \mu_2)$ of the initial target state, and search all tuples $(\bar{\mu}_1, \bar{\mu}_2) \neq (\mu_1, \mu_2)$ and $(\bar{\lambda}_1, ..., \bar{\lambda}_d) \neq (\lambda_1, ..., \lambda_d)$ such that the $2d$-tuple $(\bar{\mu}_1 \bar{\lambda}_1, \dots \bar{\mu}_1 \bar{\lambda}_d, \bar{\mu}_2 \bar{\lambda}_1, ..., \bar{\mu}_2 \bar{\lambda}_d)$ corresponds to a non-trivial permutation of the initial $2d$-tuple $(\mu_1 \lambda_1, \dots, \mu_1 \lambda_d, \mu_2 \lambda_1, ..., \mu_2 \lambda_d)$. The only *a priori* constraint on this permutation is that it should match the greatest and smallest elements of both sets, i.e.

$$\mu_1 \lambda_1 = \bar{\mu}_1 \bar{\lambda}_1 \text{ and } \mu_2 \lambda_d = \bar{\mu}_2 \bar{\lambda}_d \ . \tag{59}$$

For the others, we have to find a permutation, $\pi \in S_{2d-2}$, such that the chain of equations

$$
\begin{array}{ccccccc}
\mu_1 \lambda_1 & & \mu_1 \lambda_d & \mu_2 \lambda_1 & & \mu_2 \lambda_d \\
\| & \cdots & \| & \| & \cdots & \| \\
\bar{\mu}_1 \bar{\lambda}_1 & & \pi(\bar{\mu}_1 \bar{\lambda}_d) & \pi(\bar{\mu}_2 \bar{\lambda}_1) & & \bar{\mu}_2 \bar{\lambda}_d
\end{array}
\tag{60}
$$

has a non-trivial solution.

In the next subsections, we characterize the transformations on the two-qubit target state that can be achieved in this setting. Because they lead to highly different results, we treat separately the case where $d$ is even and the case where $d$ is odd.

### 5.1.1 Characterization of the non-trivial transformations for even $d$

For any even $d$, it is always possible to consider a two-qudit auxiliary state $|\lambda\rangle$, that is the tensor product of a two-qubit state $|\lambda_1\rangle$ and a state $|\lambda_2\rangle \in \mathbb{C}^{d/2} \otimes \mathbb{C}^{d/2}$ (if $d = 2$, the state $|\lambda_1\rangle$ is simply $|\lambda\rangle$ and there is no state $|\lambda_2\rangle$). Using the LU operation to implement a SWAP between the two-qubit states, $|\mu\rangle$ and $|\lambda_1\rangle$, and the identity in the other dimensions (if any), we see that LU operations allow for an arbitrary transformation of the initial two-qubit target state $|\mu\rangle$:

$$|\mu\rangle \otimes (|\lambda_1\rangle \otimes |\lambda_2\rangle) \rightarrow_{LU} |\lambda_1\rangle \otimes (|\mu\rangle \otimes |\lambda_2\rangle) \ . \tag{61}$$

Note that such a transformation is a particular case of the "sub-SWAP" transformation introduced in the multipartite case (see Fig. 2). Moreover, although we presented here a transformation involving biseparable auxiliary states $|\lambda\rangle$ and $|\bar{\lambda}\rangle$, we could equivalently consider a transformation involving an auxiliary state for which the states of the 2-dimensional and $(d/2)$-dimensional sub-spaces have been previously (and subsequently) entangled using LU operations acting on the $d$-level subspace only. Adding these extra LUs to the permutation realizing the sub-SWAP yields a less obvious LU transformation. For the case $d_\mu = 2, d_\lambda = 4$, by considering all valid permutations, it is easy to see that sub-SWAP solutions are (up to LU) the only solutions.

### 5.1.2 Characterization of the non-trivial transformations for odd $d$

If $d$ is odd, the previous construction cannot be applied. We therefore expect more constraints on the possible transformations, and, in this case, it is unlikely that we can achieve an arbitrary transformation of the 2-qubit target state $|\mu\rangle$. From now on, we characterize the initial two-qubit target state by the ratio $a = \frac{\mu_2}{\mu_1} \in (0,1]$. Similarly, the ratio $\bar{a} = \frac{\bar{\mu}_2}{\bar{\mu}_1}$ characterizes the final two-qubit target state.

As already mentioned, we search here for transformations that transform the state $|\mu\rangle$ into an LU-inequivalent state $|\bar{\mu}\rangle$, i.e. with $\bar{a} \neq a$. If $|\mu\rangle$ is a maximally entangled two-qubit state, then $a = 1$ and all the Schmidt coefficients of $|\mu\rangle \otimes |\lambda\rangle$ have an even degeneracy. If $|\bar{\mu}\rangle$ is not maximally entangled, however, its Schmidt coefficients are distinct and those of $|\bar{\mu}\rangle \otimes |\bar{\lambda}\rangle$ cannot all have an even degeneracy since $d$ is odd. As a consequence, when starting with a maximally entangled 2-qubit state, we can only achieve a trivial transformation. This is why we exclude in the following the case $a = 1$. This is the first constraint resulting from the fact that $d$ is odd.

As the transformation is reversible, we can focus on transformations with $a > \bar{a}$, which correspond to decreasing the entanglement of the 2-qubit target state after the LU operation. The transformations of the 2-qubit target state that can be achieved within this context are characterized in the following theorem.

**Theorem 6.** *Let* $|\mu\rangle, |\bar{\mu}\rangle \in \mathbb{C}^2 \otimes \mathbb{C}^2$ *be 2-qubit states with sets of Schmidt coefficients respectively given by* $\left(\frac{1}{1+a}, \frac{a}{1+a}\right)$ *and* $\left(\frac{1}{1+\bar{a}}, \frac{\bar{a}}{1+\bar{a}}\right)$*, with* $a, \bar{a} \in (0,1)$ *such that* $a > \bar{a}$*. Given an odd number* $d \geq 3$*, the following two statements are equivalent:*

(i) *There exist states* $|\lambda\rangle, |\bar{\lambda}\rangle \in \mathbb{C}^d \otimes \mathbb{C}^d$ *such that* $|\mu\rangle \otimes |\lambda\rangle \to_{LU} |\bar{\mu}\rangle \otimes |\bar{\lambda}\rangle$*.*

(ii) $\bar{a} = a^{d_1/d_2}$ *for two odd numbers* $d_1, d_2 \in \mathbb{N}$ *satisfying* $d \geq d_1 > d_2 \geq 1$*.*

*Proof. (only if)* If an LU transformation is possible, then there necessarily exists a permutation $\pi$ relating the Schmidt coefficients of the initial and final product states as shown in Eq. (60). We begin by using the upper line in Eq. (60) to compute the $d$ ratios $\frac{\mu_2 \lambda_i}{\mu_1 \lambda_i} = a$ for all $i \in \{1, \dots, d\}$ and write equalities with the corresponding ratios from the bottom line. We obtain a set of $d$ equations of the form

$$a = \frac{\bar{\lambda}_{x_i}}{\bar{\lambda}_{y_i}} \bar{a}^{k_i}, \ \forall \, i = 1, \dots, d \,, \tag{62}$$

with $k_i \in \{-1, 0, 1\}$ and $x_i, y_i \in \{1, \dots, d\}$. All the non-trivial transformations can be found by solving this set of equations. Because we need some elements of the solution for later constructions, we now provide an explicit method to find the non-trivial solutions of these equations.

Because the ends of the chain of equalities (60) are fixed, we know that $y_1 = 1$ and $x_d = d$, and that $k_1, k_d \neq -1$. Moreover, as we consider sorted Schmidt coefficients, the ratios $\bar{\lambda}_{x_1}/\bar{\lambda}_1$ and $\bar{\lambda}_d/\bar{\lambda}_{y_d}$ are at most equal to 1. Therefore, because we assume $a > \bar{a}$, we must in fact have $k_1 = k_d = 0$. We must also have $x_i \neq y_i$ for all values of $i$, as otherwise it would imply $a = (\bar{a})^{k_i}$, $k_i \in \{-1, 0, 1\}$, which necessarily leads to a trivial solution with $a = \bar{a}$. Consequently, in the $d$ equations, each variable $\bar{\lambda}_i$ ($i \in \{1, \dots, d\}$) must appear precisely twice, in two different equations. We now describe a method to eliminate all these variables, yielding the relation between $a$ and $\bar{a}$ stated in the theorem.

We start by selecting the two equations containing $\bar{\lambda}_1$. If $\bar{\lambda}_1$ appears as a numerator in one equation and as a denominator in the other, we multiply these equations side by side and

replace the two initial equations by the resulting equation. In this way, the resulting set of $d-1$ equations does not contain the variable $\bar{\lambda}_1$ anymore. If $\bar{\lambda}_1$ appears in both equations as a numerator or as a denominator, we invert both sides of one of these equations and proceed as explained above to get a set of $d-1$ equations that do not involve the variable $\bar{\lambda}_1$. Repeating this process at most $d-1$ times[8], we can eliminate all the $\bar{\lambda}_i$ variables, yielding an equality of the general form

$$a^{d-2r} = (\bar{a})^{\sum_{i=1}^{d} \pm k_i} , \tag{63}$$

where $r \geq 0$ is a integer related to the effective number of equation inversions that have been performed. If there is no inversion, the exponent of $a$ is $d$. It is otherwise decreased by 2 for each inversion, and the corresponding exponent $k_i$ in the right-hand side gets a minus sign.

The exponent of $a$ is obviously odd and at most equal to $d$. We now show that the exponent associated to $\bar{a}$ has to be odd as well. Any exponent $k_i = 0$ stems originally from the quotient of two $\bar{\mu}_1$ or two $\bar{\mu}_2$ when extracting Eqs. (62) from the chain (60). Because $\bar{\mu}_1$ and $\bar{\mu}_2$ both appear precisely $d$ times in these equations and the other exponents $k_i = \pm 1$ consume exactly one $\bar{\mu}_1$ and one $\bar{\mu}_2$, exponents $k_i = 0$ have to come in pairs (one corresponding to $\bar{\mu}_1/\bar{\mu}_1$ and the other one to $\bar{\mu}_2/\bar{\mu}_2$). Therefore, $\sum_{i=1}^{d} \pm k_i$ is a sum of an odd number of 1 or $-1$, which is always an odd integer. Furthermore, since we have $k_1 = k_d = 0$, this sum is at most equal to $d-2$. As a consequence, we must have $a^{d_1} = (\bar{a})^{d_2}$ with $d_1 \leq d$ and $d_2 \leq d-2$ two odd integers. If $d_1$ and $d_2$ have different signs, then $a^{|d_1|}(\bar{a})^{|d_2|} = 1$, which for $a, \bar{a} \in (0,1)$ leads to a contradiction. We can thus consider them both to be positive and, because we consider transformations with $a > \bar{a}$, we have $d_1 > d_2$. This concludes the proof of the necessary condition.

*(if)* To prove the sufficient condition, we constructively show how to build states $|\lambda\rangle$ and $|\bar{\lambda}\rangle$ enabling the transformation $|\mu\rangle \otimes |\lambda\rangle \to_{LU} |\bar{\mu}\rangle \otimes |\bar{\lambda}\rangle$, with $\bar{a} = a^{d_1/d_2}$, for any odd $d_1$ and $d_2$ satisfying $d \geq d_1 > d_2 \geq 1$. We divide the proof into the following two cases: *(i)* $d_1 = d$ and *(ii)* $d_1 < d$.

*(i)* Writing $b = a^{\frac{d}{d_2}-1}$, the unnormalized sets of Schmidt coefficients of $|\mu\rangle$ and $|\bar{\mu}\rangle$ read $\mu = \{1, a\}$ and $\bar{\mu} = \{1, ab\}$, respectively[9]. For the Schmidt coefficients of $|\lambda\rangle$ and $|\bar{\lambda}\rangle$, we choose the sets

$$\{\lambda_i\} = \{a^i\}_{\text{even } i=0}^{d-d_2-2} \cup \{a^i b\}_{\text{even } i=2}^{d-d_2-2} \cup \{b^i\}_{i=1}^{d_2+1} , \tag{64}$$

and

$$\{\bar{\lambda}_i\} = \{a^i\}_{i=0}^{d-d_2-1} \cup \{b^i\}_{i=1}^{d_2} , \tag{65}$$

respectively. To show that these sets correspond to a valid LU transformation, we must show that the tensor product $\mu \otimes \lambda$ gives the same set as $\bar{\mu} \otimes \bar{\lambda}$. These tensor products read respectively

$$\{a^i\}_{i=0}^{d-d_2-1} \cup \{a^i b\}_{i=2}^{d-d_2-1} \cup \{b^i\}_{i=1}^{d_2+1} \cup \{a b^i\}_{i=1}^{d_2+1} , \tag{66}$$

and

$$\{a^i\}_{i=0}^{d-d_2-1} \cup \{a^i b\}_{i=1}^{d-d_2} \cup \{b^i\}_{i=1}^{d_2} \cup \{a b^i\}_{i=2}^{d_2+1} . \tag{67}$$

---

[8]For some configurations of the $\bar{\lambda}_i$ variables, two variables could be eliminated in a single step (as is always the case for the last step), yielding an equality between some power of $a$ and some power of $\bar{a}$. If there are still some $\bar{\lambda}_i$ variables to eliminate, another relation between $a$ and $\bar{a}$ can be obtained by following the same procedure. In such case, the system of equations only has a non-trivial solution if all the relations between $a$ and $\bar{a}$ are equivalent. Multiplying them all side by side, we obtain an equation that has the same form (see Eq. (63)) as in the general case.

[9]Note that we use here set notations instead of tuples for convenience. Some elements in these sets might however be degenerate.

The only difference between these sets is that the first one contains the element $b^{d_2+1}$ (in its third subset) while, in the second set, this is replaced by $a^{d-d_2}b$ (in the second subset). However, since $a^{d-d_2} = b^{d_2}$, these elements are in fact equal. This concludes the proof of case $(i)$. It should be stressed here that the solution we built for $|\lambda\rangle$ and $|\bar{\lambda}\rangle$ is not necessarily the only solution allowing a transformation from $|\mu\rangle$ to $|\bar{\mu}\rangle$. The idea behind this construction and how to build other solutions will be explained in more details in the examples following the proof of the theorem.

$(ii)$ If $d_1$ does not take the maximal value, we show that we can build a solution using a solution from case $(i)$ for a lower dimension. Indeed, as both $d$ and $d_1$ must be odd, the condition $d_1 < d$ implies that there exists an integer $k > 0$ such that $d_1 + 2k = d$. We can then divide the $d$-tuple of Schmidt coefficients of $|\lambda\rangle$ into $k$ 2-tuples $\lambda^i_{(2)}$ and one $d_1$-tuple $\lambda_{(d_1)}$. In $\lambda_{(d_1)}$, we choose Schmidt coefficients of an auxiliary state allowing a transformation from the initial 2-qubit state $|\mu\rangle$ to the final 2-qubit state $|\bar{\mu}\rangle$, which has $\bar{a} = a^{d_1/d_2}$. From case $(i)$, we know that this can indeed be achieved for any odd $d_2$ satisfying $1 \leq d_2 < d_1$. For the $k$ 2-tuples $\lambda^i_{(2)}$, we simply choose Schmidt coefficients corresponding to the final 2-qubit state $|\bar{\mu}\rangle$, i.e. $\lambda^i_{(2)} = \bar{\mu}$, $\forall i = 1, \ldots, k$.

Using an LU that, in the corresponding subspaces, has the effect of swapping each 2-tuple $\lambda^i_{(2)}$ with the 2-tuple $\mu$ of the initial 2-qubit state, and performs the non-trivial transformation from case $(i)$ in the $d_1$-dimensional subspace, we achieve a transformation that has the desired final 2-qubit state $|\bar{\mu}\rangle$. Regarding the final auxiliary state, $\bar{\lambda}$ has the same structure as $\lambda$, but with $\bar{\lambda}^i_{(2)} = \mu$, $\forall i = 1, \ldots, k$, and $\bar{\lambda}_{(d_1)}$ corresponding to the $d_1$-tuple of Schmidt coefficients of the final auxiliary state of the transformation performed in the $d_1$-dimensional subspace.

This concludes the proof of case $(ii)$, and with it the proof of the sufficient part of the theorem. $\qquad\square$

The construction used to solve the case $d_1 < d$ in the sufficient part of the proof is a useful tool allowing one to embed a known solution into a larger space. Because this type of solution consists in dividing the $d$-level space into some direct sum of different subspaces, we call these solutions "direct-sum solutions" (see Fig. 3). We detail now the idea behind the constructive proof given above for the other case ($d_1 = d$) and, through explicit examples, illustrate the fact that several auxiliary states can be used for a given 2-qubit state transformation.

Theorem 6 shows that for any non-trivial transformation we can express $\bar{a}$ as a power of $a$. As a consequence, the ratios of Schmidt coefficients appearing in Eqs. (62) correspond also to some powers of $a$. This suggests that, up to some normalization factor, we can express the Schmidt coefficients themselves as powers of $a$. In this sense, Eqs. (62) characterize the "multiplicative gaps", in terms of power of $a$, between couples of Schmidt coefficients $(\bar{\lambda}_{x_i}, \bar{\lambda}_{x_j})$. Because the parameter $k_i$ in these equations can only take three different values, we have only three possible gaps. From the relation $\bar{a} = a^{d/d_2}$ (remember that we assume here $d_1 = d$), we obtain the following explicit expressions for these gaps:

$$\begin{aligned}
&\text{If } k_i = -1, && \frac{\bar{\lambda}_{x_i}}{\bar{\lambda}_{y_i}} = a^{\frac{d}{d_2}+1} \equiv g_{++}\,, \\
&\text{If } k_i = 0, && \frac{\bar{\lambda}_{x_i}}{\bar{\lambda}_{y_i}} = a \equiv g_+\,, \\
&\text{If } k_i = 1, && \frac{\bar{\lambda}_{x_i}}{\bar{\lambda}_{y_i}} = a^{1-\frac{d}{d_2}} \equiv g_-\,.
\end{aligned} \qquad (68)$$

The gaps $g_{++}$ and $g_+$ correspond to a positive power of $a$ ($g_{++}$ to a greater power of $a$ than $g_+$), while $g_-$ corresponds to a negative power of $a$.

In the case of a transformation with $d_1$ taking the maximal value $d$, the parameter $r$ in Eq. (63) has to be zero (there is no equation inversion to perform) and we have $\sum_{i=2}^{d-1} k_i = d_2$.

In this case, it is easy to see that the product of the positive gaps is precisely equal to the inverse of the product of the negative gaps. As a consequence, we can use these gaps to arrange the Schmidt coefficients of the auxiliary state $|\bar{\lambda}\rangle$ in closed cycles (see for instance Fig. 4). To build such a cycle, one starts with the largest Schmidt coefficient, i.e. $\bar{\lambda}_1$, and select the equation from the list (62) in which $\bar{\lambda}_1$ appears in the denominator. The Schmidt coefficient appearing in the numerator in this equation, say $\bar{\lambda}_x$, is then equal to $\bar{\lambda}_1$ multiplied by some gap. Since $\bar{\lambda}_1$ is the greatest Schmidt coefficient, this gap has to be positive. As there is no equation inversion when $d_1 = d$, $\bar{\lambda}_x$ has to appear in the denominator of some other equation, which can be used to relate $\bar{\lambda}_x$ to another Schmidt coefficient via a positive gap. Continuing to follow this list of equations, we arrive eventually at the equation in which $\bar{\lambda}_1$ appears in the numerator. Again, because $\bar{\lambda}_1$ is the greatest Schmidt coefficient, this equation is necessarily associated to a negative gap which closes the cycle. If there are still equations left in the list, we start another cycle of Schmidt coefficients. Note that in the case of multiple cycles, each cycle must produce the same relation between $a$ and $\bar{a}$, as we otherwise have $a = \bar{a} = 1$. As we illustrate now for $d_1 = d = 5$ and $d_2 = 1$, this can be used to give a schematic picture of all transformations turning the initial (unnormalized) 2-qubit Schmidt vector $(1, a)$ into $(1, a^5)$.

In this case, the three possible gaps given in Eq. (68) read

$$
\begin{aligned}
g_{++} &= a^6 \, , \\
g_{+} &= a \, , \\
g_{-} &= a^{-4} \, ,
\end{aligned}
\tag{69}
$$

and there are, up to reordering, only two sets $\{k_i\}_{i=1}^5$ such that $\sum_{i=2}^4 k_i = d_2 = 1$ (recall that we always have $k_1, k_5 = 0$):

$$
k_a = \{0, 1, 0, 0, 0\}, \quad k_b = \{0, 1, 1, -1, 0\} \, .
\tag{70}
$$

Let us first consider the case $k_a = \{0, 1, 0, 0, 0\}$. In this case, there is no gap $g_{++}$ and only one gap $g_-$. As a consequence, the only cycle that we can create starts with the largest Schmidt coefficient $\lambda_1$, then uses all the four positive gaps to go through the remaining four Schmidt coefficients and closes the cycle using the negative gap (see Fig. 4).

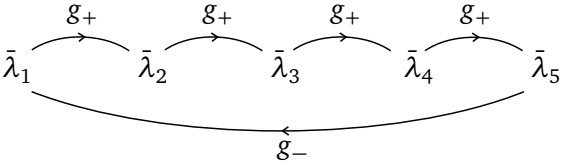

Figure 4: Cycle associated with $k_a = (0, 1, 0, 0, 0)$

All Schmidt coefficients are expressed as a function of $\bar{\lambda}_1$ (which accounts for the normalization). Setting it to 1, the corresponding (unnormalized) Schmidt vector for $|\bar{\lambda}\rangle$ reads

$$
\bar{\lambda} = (1, a, a^2, a^3, a^4) \, .
\tag{71}
$$

The Schmidt vector of $|\lambda\rangle$ can be deduced by considering the equation $\mu \otimes \lambda \sim \bar{\mu} \otimes \bar{\lambda}$. In summary, using (unnormalized) Schmidt vectors to denote the bipartite states, we have the transformation:

$$
(1, a) \otimes (1, a^2, a^4, a^6, a^8) \to_{LU} (1, a^5) \otimes (1, a, a^2, a^3, a^4) \, .
\tag{72}
$$

We now turn to the second case $k_b = \{0, 1, 1, -1, 0\}$. Because we have here the two types of positive gaps, $g_+$ and $g_{++}$, we can build several cycles, see for instance Figs. 5 and 6. However,

not every cycle corresponds to a valid non-trivial transformation. Indeed, inverting the gaps of the cycle in Fig. 5 to get equations of the form (62), we get:

$$a = \frac{\bar{\lambda}_3}{\bar{\lambda}_1}\,\bar{a}^{-1} = \frac{\bar{\lambda}_1}{\bar{\lambda}_2}\,\bar{a} = \frac{\bar{\lambda}_4}{\bar{\lambda}_3} = \frac{\bar{\lambda}_5}{\bar{\lambda}_4} = \frac{\bar{\lambda}_2}{\bar{\lambda}_5}\,\bar{a}\,. \tag{73}$$

These equations are compatible with the relation $a^5 = \bar{a}$ but, writing explicitly $\bar{a} = \frac{\bar{\mu}_2}{\bar{\mu}_1}$, we see that the Schmidt coefficient $\bar{\lambda}_1\bar{\mu}_2$ appears twice in this set of equations, whereas $\bar{\lambda}_1\bar{\mu}_1$ and $\bar{\lambda}_1\bar{\mu}_2$ should both occur precisely once. As a consequence, these equations have a solution only if $\bar{\mu}_1 = \bar{\mu}_2$, showing that this cycle corresponds to a trivial transformation with $a = \bar{a} = 1$.

The cycle depicted in Fig. 6 corresponds to the only non-trivial transformation in the case $k_b = \{0, 1, 1, -1, 0\}$. Indeed, as noted in the proof of Theorem 6, a non-trivial transformation necessarily implies $a = \frac{\bar{\lambda}_{x_1}}{\bar{\lambda}_1} = \frac{\bar{\lambda}_d}{\bar{\lambda}_{y_d}}$ for some $x_1 \in \{2, \dots, d\}$ and $y_d \in \{1, \dots, d-1\}$. As we consider an unnormalized Schmidt vector, we can without loss of generality set $\bar{\lambda}_1 = 1$. This implies $\bar{\lambda}_{x_1} = a$. As, in this case, all gaps correspond to integer powers of $a$ (which is not always the case as we illustrate later), and we can here only have a single cycle (as $d_1$ has the largest possible value), there cannot be another Schmidt coefficient between $\bar{\lambda}_1$ and $\bar{\lambda}_{x_1}$. We thus have $x_1 = 2$ and $\bar{\lambda}_2 = a$. For a similar reason we must have $\frac{\bar{\lambda}_5}{\bar{\lambda}_4} = a$. With these two constraints, it follows that the cycle in Fig. 6 is the only possible solution. It corresponds to the transformation

$$(1, a) \otimes (1, a^4, a^6, a^8, a^{12}) \to_{LU} (1, a^5) \otimes (1, a, a^4, a^7, a^8)\,. \tag{74}$$

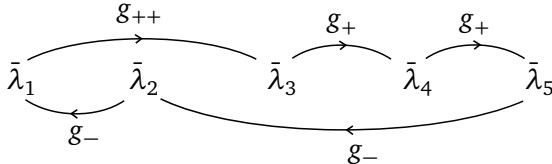

Figure 5: Example of cycle associated with $k_b = \{0, 1, 1, -1, 0\}$ that leads to a trivial transformation with $a = \bar{a} = 1$.

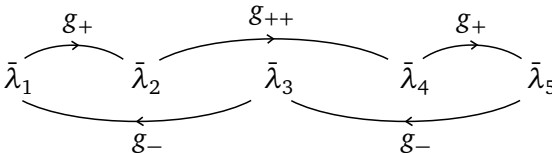

Figure 6: This cycle is the only one leading to a valid LU transformation in the case $k_b = \{0, 1, 1, -1, 0\}$.

For a given qubit transformation from $|\mu\rangle$ to $|\bar{\mu}\rangle$, there may be several choices of (odd dimensional) states $|\lambda\rangle$ and $|\bar{\lambda}\rangle$, each transformation corresponding to a specific unitary operation. When $d_1 = d$, each solution corresponds to a specific cycle, and there are only finitely many possibilities. As $d$ increases, however, the length of potential cycles increases, leading to more possible cycles, and thus more transformations. For example, in the case of $d = d_1 = 7$

and $d_2 = 3$, we have the three distinct (not normalised) transformations:

$$(1, a) \otimes (1, a^{4/3}, a^{8/3}, a^{10/3}, a^4, a^{16/3}, a^{20/3})$$
$$\rightarrow_{LU} (1, a^{7/3}) \otimes (1, a, a^{4/3}, a^{8/3}, a^4, a^{13/3}, a^{16/3}), \tag{75}$$

$$(1, a) \otimes (1, a^{2/3}, a^{4/3}, a^2, a^{8/3}, a^{10/3}, a^4)$$
$$\rightarrow_{LU} (1, a^{7/3}) \otimes (1, a^{2/3}, a, a^{4/3}, a^{5/3}, a^2, a^{8/3}), \tag{76}$$

$$(1, a) \otimes (1, a^{4/3}, a^2, a^{8/3}, a^{10/3}, a^4, a^{16/3})$$
$$\rightarrow_{LU} (1, a^{7/3}) \otimes (1, a, a^{4/3}, a^2, a^{8/3}, a^3, a^4). \tag{77}$$

Investigating which cycle corresponds to a non-trivial solution becomes more involved as $d$ increases. For $d > 5$, the cycles do not necessarily consist of a sequence containing all the positive gaps, followed by a sequence containing all the negative gaps (as in Figs. 4 and 6). More intricate cycle structures appear, such as for instance the cycle corresponding to transformation in Eq. (76) (see Fig. 7).

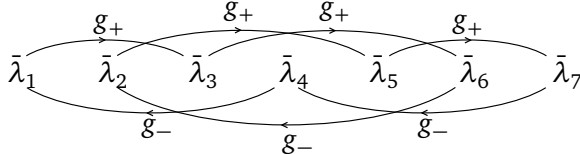

Figure 7: Cycle associated with the transformation in Eq. (76).

This last example concludes our illustration of the possible transformations of 2-qubit states under multi-state LUs characterized in Theorem 6. In the next section we address the possible transformations of bipartite states of higher dimension.

## 5.2 Non-trivial solutions in higher dimensional non-homogeneous systems

We show here that non-trivial transformations are not only possible when one of the initial states is a 2-qubit state but also occur in all non-homogeneous systems, i.e. systems with $d_\mu \neq d_\lambda$.

To begin, let us look at the non-trivial transformations in the case $d_\mu = 3$, $d_\lambda = 4$. By considering all valid permutations, one can show that there are four non-trivial, non-direct-sum solutions.

$$(1, a, a^2) \otimes (1, a^3, a^6, a^9) \sim (1, a^4, a^8) \otimes (1, a, a^2, a^3) \tag{78}$$

$$(1, a^2, a^4) \otimes (1, a^3, a^5, a^6) \sim (1, a^4, a^5) \otimes (1, a^2, a^3, a^5) \tag{79}$$

$$(1, a, a^5) \otimes (1, a^3, a^5, a^6) \sim (1, a^4, a^5) \otimes (1, a, a^3, a^6) \tag{80}$$

$$(1, a, a^5) \otimes (1, a^2, a^3, a^5) \sim (1, a^2, a^4) \otimes (1, a, a^3, a^6). \tag{81}$$

Note, perhaps surprisingly in this higher dimensional $d_\mu = 3$, $d_\lambda = 4$ case, all non-trivial, non-direct-sum solutions are still characterised by a single variable. However, as the dimensions grow, the number of non-trivial solutions will also grow. This makes further investigation of this feature challenging. Nonetheless, we can generally observe that for any non-homogeneous system ($d_\mu \neq d_\lambda$), there is always at least one non-trivial solution. As we illustrate in the following observation, it is indeed possible to generalize the transformation given in Eq. (72) to pairs of bipartite states of arbitrary (but different) dimensions.

**Observation 7.** *For any $d_\mu, d_\lambda \geq 2$, with $d_\mu < d_\lambda$, the tuples of Schmidt coefficients*

$$\mu = \left(1, a^{d_\lambda}, a^{2d_\lambda}, \ldots, a^{(d_\mu-1)d_\lambda}\right), \tag{82}$$

$$\lambda = \left(1, a, a^2, \ldots, a^{d_\lambda-1}\right), \tag{83}$$

*and*

$$\bar{\mu} = \left(1, a, a^2, \ldots, a^{d_\mu-1}\right), \tag{84}$$

$$\bar{\lambda} = \left(1, a^{d_\mu}, a^{2d_\mu}, \ldots, a^{(d_\lambda-1)d_\mu}\right), \tag{85}$$

*lead to the non-trivial transformation $|\mu\rangle \otimes |\lambda\rangle \to_{LU} |\bar{\mu}\rangle \otimes |\bar{\lambda}\rangle$ (see Fig. 3).*

This can easily be verified by computing the tensor products $\mu \otimes \lambda$, and $\bar{\mu} \otimes \bar{\lambda}$ and checking that they are equal up to reordering. We now move on to homogeneous systems.

### 5.3 Homogeneous Systems

As we showed in the previous section, non-trivial transformations can be found in all non-homogeneous systems. However, the general example shown in the previous section cannot be used in homogeneous systems. This is because, when $d_\mu = d_\lambda$, the transformation in Observation 7 corresponds to SWAP. Consequently, we must look for alternate non-trivial transformations.

Building on the non-trivial transformations characterized in the previous section, we first show that non-trivial transformations can be found in all homogeneous systems with $d \geq 4$. As an application, we then use such transformations to show that the source entanglement, an entanglement measure that was defined in Ref. [45], is not an additive measure for bipartite pure states. Finally, we prove that there does not exist any non-trivial transformations for the two remaining homogeneous systems ($d = 2, 3$). In this last part, we make use of the elementary symmetric polynomials approach presented in the introductory part of Section 5.

#### 5.3.1 Building non-trivial transformations

We show here the following observation:

**Observation 8.** *There exists at least one non-trivial transformation in all homogeneous systems with $d_\mu = d_\lambda \geq 4$.*

We proceed by first demonstrating non-trivial solutions for all non-prime dimensions. Then we demonstrate a solution for all odd (and therefore all prime) dimensions greater than $d = 5$. Finally, we provide an explicit solution for $d = 5$.

If the dimension $d_\mu = d_\lambda = d$ is not a prime number, then it can always be factorized into two smaller dimensions, $d_1$ and $d_2$, with $d = d_1 d_2$. It is therefore possible to consider, as initial states, two product states $|\lambda\rangle = |\lambda_{d_1}\rangle \otimes |\lambda_{d_2}\rangle$ and $|\mu\rangle = |\mu_{d_1}\rangle \otimes |\mu_{d_2}\rangle$. We can then obviously get a non-trivial transformation if we use local unitaries to swap only the states corresponding to the $d_1-$ (or $d_2-$) dimensional subspace, yielding a transformation of the form

$$\left(|\mu_{d_1}\rangle \otimes |\mu_{d_2}\rangle\right) \otimes \left(|\lambda_{d_1}\rangle \otimes |\lambda_{d_2}\rangle\right) \xrightarrow{LU} \left(|\lambda_{d_1}\rangle \otimes |\mu_{d_2}\rangle\right) \otimes \left(|\mu_{d_1}\rangle \otimes |\lambda_{d_2}\rangle\right). \tag{86}$$

Note that, for simplicity, we present here again a transformation with bipartite states that have a tensor product structure across their $d$ dimensions. Using local unitaries, we could also create entanglement across these $d_1$- and $d_2$-dimensional subspaces to provide a similar transformation involving only fully-entangled states.

If $d$ is prime, we obviously cannot have a non-trivial solution of this form. However, instead of decomposing the dimension into a product, we can decompose it into a sum. If $d$ is large enough, this sum decomposition can lead to non-trivial solutions. We demonstrate this with an example. In the case $d = 7$, we can split the 7 Schmidt coefficients of both initial states into one set of 4 Schmidt coefficient and one set of 3 Schmidt coefficients. Splitting further the sets of 4 Schmidt coefficients into a tensor product of two sets of 2 Schmidt coefficients, we take for $\lambda$ (and similarly for $\mu$) the following structure:

$$\lambda = c\left(\lambda^1_{(2)} \otimes \lambda^2_{(2)}\right) \oplus (1-c)\lambda_{(3)} \,, \tag{87}$$

where $\lambda^1_{(2)}$ and $\lambda^2_{(2)}$ are tuples of Schmidt coefficients of 2-qubit states, $\lambda_{(3)}$ is a tuple of Schmidt coefficients of a 2-qutrit state and $0 < c < 1$ is a mixing parameter.

To achieve a non-trivial transformation from this structure, we exploit a non-trivial transformation in $(\mathbb{C}^2 \otimes \mathbb{C}^3)^{\otimes 2}$ that can be deduced from Theorem 6. For this system, the theorem shows that there exists a non-trivial transformation of the 2-qubit target state corresponding to a final state with $\bar{a} = a^3$. In term of Schmidt coefficients, this transformation reads

$$\frac{1}{1+a}(1,a) \otimes \frac{1}{1+a^2+a^4}(1,a^2,a^4) \sim \frac{1}{1+a^3}(1,a^3) \otimes \frac{1}{1+a+a^2}(1,a,a^2)\,. \tag{88}$$

As a consequence, using

$$\mu = c\left(\frac{1}{1+a}(1,a) \otimes \frac{1}{1+b}(1,b)\right) \oplus (1-c)\frac{1}{1+a+a^2}(1,a,a^2)\,, \tag{89}$$

$$\lambda = c'\left(\frac{1}{1+a^3}(1,a^3) \otimes \frac{1}{1+b'}(1,b')\right) \oplus (1-c')\frac{1}{1+a^2+a^4}(1,a^2,a^4)\,, \tag{90}$$

we can achieve a transformation to

$$\bar{\mu} = c\left(\frac{1}{1+a^3}(1,a^3) \otimes \frac{1}{1+b}(1,b)\right) \oplus (1-c)\frac{1}{1+a^2+a^4}(1,a^2,a^4)\,, \tag{91}$$

$$\bar{\lambda} = c'\left(\frac{1}{1+a}(1,a) \otimes \frac{1}{1+b'}(1,b')\right) \oplus (1-c')\frac{1}{1+a+a^2}(1,a,a^2)\,. \tag{92}$$

This argument holds for any odd dimension, $d \geq 7$. This is because for any $d \geq 7$, $d-3$ is even and at least equal to 4 (so that the corresponding subspace can be further split into a tensor product of two non-trivial subspaces). Therefore we can construct a state $\lambda = c(\lambda_{(2)} \otimes \lambda_{(d/2)}) \oplus (1-c)\lambda_{(3)}$ (where, as before, the subscript indicates the dimension of the corresponding tuple). Then we simply apply the same type of transformation as in the example. As this argument holds for any odd dimension $d \geq 7$, it holds in particular for all prime dimensions greater than five.

Finally, in the case of $d = 5$, we provide the following explicit example of a non-trivial transformation:

$$(1,a,a^{4/3},a^2,a^{8/3}) \otimes (1,a^{1/3},a^{2/3},a,a^{4/3})$$
$$\sim (1,a^{1/3},a^{4/3},a^{5/3},a^2) \otimes (1,a^{2/3},a,a^{4/3},a^2)\,. \tag{93}$$

This concludes the proof of the observation that in all homogeneous bipartite systems (except those of dimension 2 and 3), there are non-trivial multi-state LU transformations.

Before completing the last remaining cases of $d = 2,3$ in Section 5.3.3, as an application of the transformations we described in this section, we first show that the source entanglement [45], $E_s$, is not an additive measure of entanglement for bipartite states, in contrast, for instance, to the Von Neumann entropy (of pure states).

### 5.3.2 Source Entanglement of Bipartite Systems

The source entanglement is a measure of entanglement, ranging from 0 to 1, which measures how difficult it is to reach a state using LOCC. For a bipartite state $|\lambda\rangle \in \mathbb{C}^d \otimes \mathbb{C}^d$, with set of Schmidt coefficients $\lambda$, it is given by [45]

$$E_s(\lambda) = 1 - \sum_{\sigma \in S_d} \frac{\left( \sum_{k=1}^{d} \sigma(k)\lambda_k \right)^{d-1}}{\prod_{k=1}^{d-1} \left( \sigma(k) - \sigma(k+1) \right)}, \tag{94}$$

where the sum runs over all permutations $\sigma$ from the permutation group of $d$ elements, $S_d$.

**Observation 9.** *The source entanglement is not an additive measure of entanglement for bipartite states*

Consider a transformation involving the states given in Eqs. (89)-(92). Obviously, we have $E_s(\mu \otimes \lambda) = E_s(\bar{\mu} \otimes \bar{\lambda})$. However, for some values of the parameters of the transformation, we can have $E_s(\lambda) + E_s(\mu) < E_s(\bar{\lambda}) + E_s(\bar{\mu})$. In that case, the source entanglement can increase more in the transformation from $|\lambda\rangle$ to $|\bar{\lambda}\rangle$ than it decreases in the transformation from $|\mu\rangle$ to $|\bar{\mu}\rangle$. To give an example, choosing the parameters $a = 0.3$, $b = 0.01$, $c = 0.01$, $b' = 0.3$ and $c' = 0.8$, this difference amounts to $\left( E_s(\bar{\lambda}) + E_s(\bar{\mu}) \right) - \left( E_s(\lambda) + E_s(\mu) \right) = 0.56$. Using a state $|\mu\rangle$ that is easy to reach ($E_s(\mu) = 0.005$), we can transform an easily reachable state $|\lambda\rangle$ with $E_s(\lambda) = 0.11$ into a state $|\bar{\lambda}\rangle$ with $E_s(\bar{\lambda}) = 0.68$, which is much more difficult to reach via LOCC. This once again demonstrates that multi-state LOCC transformations are much richer than their single-state counterparts.

### 5.3.3 Characterizing trivial solutions using ESPs

In this section, we show that for the two homogeneous systems for which we did not provide examples of non-trivial transformations, namely those with $d = 2, 3$, only trivial transformations are possible. As explained previously, ESPs provide a natural framework for studying bipartite multi-state LU transformations through the following necessary and sufficient conditions: the transformation $|\mu\rangle \otimes |\lambda\rangle \rightarrow_{LU} |\bar{\mu}\rangle \otimes |\bar{\lambda}\rangle$ is possible if and only if

$$e_i(\mu \otimes \lambda) = e_i(\bar{\mu} \otimes \bar{\lambda}), \forall i \in \{1, \ldots, d_\mu d_\lambda\}. \tag{95}$$

Moreover, it was demonstrated in Ref. [42] that if the Schmidt coefficients have a tensor product structure, as in the present case, then the ESPs $e_i(\mu \otimes \lambda)$ can be expressed in terms of the ESPs $s_i \equiv e_i(\lambda)$ and $t_i \equiv e_i(\mu)$, i.e. in term of the ESPs of the marginals. For example, in the case $d_\mu = d_\lambda = 2$, we have:

$$e_1(\mu \otimes \lambda) = s_1 t_1 \tag{96}$$

$$e_2(\mu \otimes \lambda) = s_1^2 t_2 + s_2 t_1^2 - 2s_2 t_2 \tag{97}$$

$$e_3(\mu \otimes \lambda) = s_1 t_1 s_2 t_2 \tag{98}$$

$$e_4(\mu \otimes \lambda) = s_2^2 t_2^2. \tag{99}$$

Although this decomposition does not usually help solving Eqs. (95), which is typically a large set of high degree polynomial equations, we now show that it is very useful to identify trivial LU transformations. A transformation, $|\mu\rangle \otimes |\lambda\rangle \rightarrow_{LU} |\bar{\mu}\rangle \otimes |\bar{\lambda}\rangle$, is trivial if the tuple of tuples $(\mu, \lambda)$ is equal up to reordering to the tuple of tuples $(\bar{\mu}, \bar{\lambda})$. This accounts indeed for both $\mathbb{1}$ and SWAP trivial transformations. Since the order of the Schmidt coefficients in each tuple does not matter, we can replace the tuples of Schmidt coefficients by their corresponding

tuples of ESPs. In the case $d_\mu = d_\lambda = 2$, using the simplified notations $s_i$ and $t_i$ (resp. $\bar{s}_i$ and $\bar{t}_i$) for the ESPs of the tuples $\mu$ and $\lambda$ ($\bar{\mu}$ and $\bar{\lambda}$), we then have a trivial transformation if and only if

$$\left((s_1, s_2), (t_1, t_2)\right) \sim \left((\bar{s}_1, \bar{s}_2), (\bar{t}_1, \bar{t}_2)\right). \tag{100}$$

As the first order ESP of any normalized tuple of Schmidt coefficients is equal to 1, for normalized states, the previous equation reduces to

$$(s_2, t_2) \sim (\bar{s}_2, \bar{t}_2). \tag{101}$$

This is a usual equivalence relation between two tuples of two variables. This equivalence holds if and only if the two ESPs of the two tuples are equal, i.e. if and only if

$$s_2 + t_2 = \bar{s}_2 + \bar{t}_2, \tag{102}$$

$$s_2 t_2 = \bar{s}_2 \bar{t}_2. \tag{103}$$

On the other hand, all the solutions for the transformation $|\lambda\rangle \otimes |\mu\rangle \to_{LU} |\bar{\lambda}\rangle \otimes |\bar{\mu}\rangle$ can be obtained by solving the equations $e_i(\lambda \otimes \mu) = e_i(\bar{\lambda} \otimes \bar{\mu})$, $\forall i = 1, \ldots, 4$. Using the decompositions in Eqs. (96) to (99), and taking into account that we consider normalized states, this set of equations is equivalent to:

$$s_2 + t_2 - 2s_2 t_2 = \bar{s}_2 + \bar{t}_2 - 2\bar{s}_2 \bar{t}_2, \tag{104}$$

$$s_2 t_2 = \bar{s}_2 \bar{t}_2. \tag{105}$$

As these two equations are equivalent to the conditions (102) and (103) for having trivial solutions, we conclude that there are only trivial transformations for $d_\mu = d_\lambda = 2$.

In order to generalize this approach for higher dimensions we present the following theorem:

**Theorem 10** (Equivalence between two tuples of tuples). *Let $\underline{s} = (s_1, s_2, \ldots, s_d) \in \mathbb{R}_+^d$ with $s_{i+1} \leq s_i$ and let $\underline{t}, \bar{\underline{s}}, \bar{\underline{t}}$ be defined similarly. Then $(\underline{s}, \underline{t})$ is equal to $(\bar{\underline{s}}, \bar{\underline{t}})$ up to reordering iff the following conditions hold:*

1. $e_i(\underline{s}) + e_i(\underline{t}) = (\bar{\ \ })$, $\forall i = 1, \ldots, d$ and

2. $\sum_{i+j=k} e_i(\underline{s}) e_j(\underline{t}) = (\bar{\ \ })$, $\forall k = 1, \ldots, 2d$

*where $(\bar{\ \ })$ indicates the same as the LHS but with all variables barred.*

The proof of this theorem is provided in the Appendix D. As $e_{i+1}(\lambda) \leq e_i(\lambda)$ for any normalized tuple of Schmidt coefficients $\lambda$, and the ESPs over $\lambda$ completely determine $|\lambda\rangle$ up to LUs, this theorem gives necessary and sufficient conditions for trivial transformations being the only possible transformations.

We now use this to show that there are only trivial solutions in the one remaining case: $d = 3$. Again, $|\lambda\rangle \otimes |\mu\rangle \to_{LU} |\bar{\lambda}\rangle \otimes |\bar{\mu}\rangle$ if and only if $e_i(\lambda \otimes \mu) = e_i(\bar{\lambda} \otimes \bar{\mu})$, $\forall i = 1, \ldots, 9$. Decomposing these equalities in term of the ESPs of the marginals, we have the set of equations:

$$s_2 + t_2 - 2s_2 t_2 = (\bar{\ \ }) \tag{106}$$

$$s_3 + t_3 + s_2 t_2 - 3(s_3 t_2 + s_2 t_3) + 3s_3 t_3 = (\bar{\ \ }) \tag{107}$$

$$s_3 t_2 + s_2 t_3 + s_2^2 t_2^2 - 2(s_3 t_2^2 + s_2^2 t_3) - s_3 t_3 = (\bar{\ \ }) \tag{108}$$

$$s_2 s_3 t_2^2 + s_2^2 t_2 t_3 - 2(s_2 s_3 t_3 + s_3 t_2 t_3) - s_2 s_3 t_2 t_3 + s_3 t_3 = (\bar{\ \ }) \tag{109}$$

$$s_3^2 t_2^3 + s_2^3 t_3^2 + s_2 s_3 t_2 t_3 - 3(s_3^2 t_2 t_3 + s_2 s_3 t_3^2) + 3s_3^2 t_3^2 = (\bar{\ \ }) \tag{110}$$

$$s_3 t_3 \left(s_3 t_2^2 + s_2^2 t_3 - 2s_3 t_3\right) = (\bar{\ \ }) \tag{111}$$

$$s_2 t_2 s_3^2 t_3^2 = (\bar{\ \ }) \tag{112}$$

$$s_3^3 t_3^3 = (\bar{\ \ }), \tag{113}$$

where again ($\bar{}$'') indicates the same as the LHS but with all the variables barred.

Now by application of Theorem 10, a transformation is trivial iff

$$s_2 + t_2 + s_3 + t_3 = (\bar{}'') \tag{114}$$

$$s_2 s_3 + t_2 t_3 = (\bar{}'') \tag{115}$$

$$(s_2 s_3) + (s_2 + s_3)(t_2 + t_3) + (t_2 t_3) = (\bar{}'') \tag{116}$$

$$(s_2 + s_3)(t_2 t_3) + (s_2 s_3)(t_2 + t_3) = (\bar{}'') \tag{117}$$

$$s_2 s_3 t_2 t_3 = (\bar{}'') . \tag{118}$$

Using the fact that $s_i, t_i > 0$, one can easily show that the set of Eqs. (106-113) implies the set of Eqs. (114-118). Therefore, the conditions for having a solution imply only trivial solutions, which shows that the only solutions in homogeneous systems with $d = 2, 3$ are the trivial $\mathbb{1}$ and SWAP transformations. This completes our analysis of multi-state bipartite LU transformations.

# 6 Conclusion

In this work, we investigated the physically motivated extension of LOCC consisting of multi-state transformations. As mentioned in the introduction, considering such an extension of LOCC is motivated by the fact that, in homogeneous systems, states are generically isolated under single-state LOCC [30], implying that the MES necessarily contains almost all states of the Hilbert space. Relaxing this setting, for instance by allowing local operations on multiple states, new transformations could be achieved and a (hopefully more practical) equivalent of the MES could be obtained.

We first showed that by performing a multi-state LOCC transformation on several multipartite pure states, it is possible to change the SLOCC class of at least one of them with only LUs. As one of the initial states could be transformed into a state that belongs to a different SLOCC class, we must, in order to characterize all possible transformations, consider potential final states from all possible SLOCC classes. As there is generically an infinite number of SLOCC classes in multipartite systems, achieving a full characterization of multipartite multi-state transformations is very unlikely. This is also one of the reasons why identifying the equivalent of the MES in the multi-state setting is probably out of reach.

In light of this first result, we focused in the present work on identifying new features of multi-state transformations of multipartite states (compared to single-state LOCC). With a 3-qubit example, we showed that a state from the MES can be reached in the multi-state setting through an LOCC transformation of two states that are not from the MES, which allows for some freedom in choosing the 2-MES. We also showed that catalytic transformations of multipartite states can be performed in the multi-state regime and that this regime can provide an advantage (in the sense of a larger success probability) for probabilistic LOCC transformations.

These results show qualitatively that multi-state LOCC allows a much larger set of possible transformations than single-state LOCC. Looking for a more systematic characterization of the new possible transformations, we also considered the simpler setting of bipartite two-state LU transformations. In this setting, we provided a full characterization of the possible transformations of a 2-qubit state, when transformed together with an arbitrary auxiliary bipartite state. We also showed that in almost all possible pairs of bipartite systems, non-trivial two-state LU transformations can be achieved.

Looking forward, our multipartite results show that multi-state LOCC has a wide range of interesting phenomena but a full characterisation is probably out of reach. Therefore, further

investigations of multipartite, multi-state transformations should either focus on the asymptotic case, or on physically relevant sets of states (for which the multi-state LOCC structure might be simpler than in the general case). Regarding the bipartite setting, the results presented here show that already LUs lead to non-trivial transformations in the multi-state setting. One could extend our work by expanding the allowed operations to LOCC. In this case, optimal protocols already exist for entanglement concentration of finitely many bipartite states [19]. However, a full characterisation seems very challenging as it would include the heavily investigated bipartite entanglement catalysis [39, 40, 43, 54] as a subset of transformations.

# Acknowledgements

We would like to thank Leonhard Czarnetzki for fruitful discussions and for his work, during his Masters thesis [58], on related problems regarding multi-state transformations. We acknowledge financial support from the SFB BeyondC (Grant No. F7107-N38), the Austrian Academy of Sciences via the Innovation Fund "Research, Science and Society", and the Austrian Science Fund (FWF) through grants P 32273-N27 and DK-ALM: W1259-N27.

# A  Changing SLOCC orbit type in the multi-state regime

In this appendix, we discuss the possibility to change the orbit type of SLOCC classes under multi-state LU transformations as in Section 4.1. We use the same notation as in the main text, i.e., the pair of states $|\psi\rangle$ and $|\bar{\psi}\rangle$ is transformed to the pair of states $|\phi\rangle$ and $|\bar{\phi}\rangle$. First, we briefly review the notion of the orbit type of SLOCC classes (states), i.e, the notion of polystable states, strictly semistable states, and states of the null-cone. Then, we show that the example presented in the main text, $|\psi\rangle = |\text{GHZ}\rangle^{\otimes 2}$, $|\bar{\psi}\rangle = |\text{W}\rangle^{\otimes 2}$, and $|\phi\rangle = |\bar{\phi}\rangle = |\text{GHZ}\rangle\,|\text{W}\rangle$ indeed constitutes an example in which a polystable state and a state from the null-cone is transformed to two states from the null-cone. We then provide additional examples showing that two strictly semistable states can be transformed to one strictly semistable and one polystable state. Furthermore, we show that two states in the null-cone can be transformed to one strictly semistable state and one state in the null-cone. Finally, we discuss that the orbit types cannot be changed arbitrarily. This is due to the fact that, obviously, the orbit type of the joint state cannot change under SLOCC. In the course of that, we discuss the orbit type of tensor products of states. To conclude, we draw a connection to SLOCC catalysis (changing SLOCC class with a catalytic SLOCC transformation).

Let us begin by reviewing the notion of the orbit type of SLOCC classes. As mentioned in the main text, states and their respective SLOCC classes can be categorized into three different types depending on geometrical properties of the orbit: polystable classes, strictly semistable classes, and the null-cone[10], see e.g. [46–49]. The orbit type plays a role in characterizing deterministic LOCC transformations within SLOCC classes. Polystable classes are those SLOCC classes that contain a critical state. Strictly semistable classes are those SLOCC classes that do not contain a critical state, but do contain a critical state in their closure. Finally, the null-cone is composed by the remaining classes, i.e., those SLOCC classes that do not contain a critical state within their closure. Not all of these types are necessarily present within a quantum system of specified local dimensions. Note that a state $|\xi\rangle$ is in the null-cone if and only if there exists a sequence of operators $S_\alpha^{(1)}, S_\alpha^{(2)}, \ldots \in \text{SL}(d, \mathbb{C})$ such that $\lim_{\alpha\to\infty} S_\alpha^{(1)} \otimes S_\alpha^{(2)} \otimes \ldots |\xi\rangle = 0$ [47]. A state $|\chi\rangle$ is strictly semistable if and only if it has

---

[10]A finer classification can be made. However, for our purposes here, the presented classification is sufficient.

the property that it is not SLOCC equivalent to any critical state, but there exists a sequence of operators as before such that $\lim_{\alpha\to\infty} S_\alpha^{(1)} \otimes S_\alpha^{(2)} \otimes \ldots |\chi\rangle \propto |\Psi_c\rangle$ for some critical state $|\Psi_c\rangle$ (with non-vanishing proportionality factor). Note also that a critical state in the closure of an SLOCC class is unique up to LUs [46,47]. Analytical and numerical methods to determine the orbit type have been devised, see e.g. [47,48,63–65].

Let us now study how the orbit type of states may change under multi-state LU transformations. Let us first consider the transformation from the main text, $|\psi\rangle = |\text{GHZ}\rangle^{\otimes 2}$, $|\bar{\psi}\rangle = |\text{W}\rangle^{\otimes 2}$, and $|\phi\rangle = |\bar{\phi}\rangle = |\text{GHZ}\rangle|\text{W}\rangle$. As explained in the main text, this transformation is possible by LUs acting jointly on the copies. Moreover, note that $|\text{GHZ}\rangle^{\otimes 2}$ is critical and thus represents a polystable SLOCC class, while $|\text{W}\rangle|\text{GHZ}\rangle$ as well as $|\text{W}\rangle^{\otimes 2}$ are in the null-cone. This can be easily seen as follows. The state $|\text{W}\rangle$ is in the null-cone, hence, there exists a sequence of operators $A_\alpha, B_\alpha, C_\alpha \in \text{SL}(2,\mathbb{C})$ such that $\lim_{\alpha\to\infty} A_\alpha \otimes B_\alpha \otimes C_\alpha |\text{W}\rangle = 0$ [47]. Then, for any state $|\zeta\rangle$, it holds that $\lim_{\alpha\to\infty}(\mathbb{1}\otimes A_\alpha)\otimes(\mathbb{1}\otimes B_\alpha)\otimes(\mathbb{1}\otimes C_\alpha)|\zeta\rangle|\text{W}\rangle = 0$, where $\mathbb{1}\otimes A_\alpha$, $\mathbb{1}\otimes B_\alpha$, $\mathbb{1}\otimes C_\alpha$ have determinant one. Thus, $|\text{GHZ}\rangle|\text{W}\rangle$ and $|\text{W}\rangle^{\otimes 2}$ are in the null-cone. Let us remark that this argument actually holds for any state in the null-cone together with an arbitrary state $|\zeta\rangle$. This shows that, indeed, it is possible to transform one state in the null-cone and one polystable state to two states that are both in the null-cone.

Similarly, it is possible to transform two strictly semistable states to one strictly semistable as well as one polystable state. Consider the transformation involving the four-partite states

$$|\psi\rangle = |\text{GHZ}\rangle^{\otimes 2} \tag{119}$$

$$|\bar{\psi}\rangle = |\chi\rangle^{\otimes 2} \tag{120}$$

$$|\phi\rangle = |\bar{\phi}\rangle = |\text{GHZ}\rangle|\chi\rangle \;, \tag{121}$$

where $|\chi\rangle = |0000\rangle + |1111\rangle + |0110\rangle + |0011\rangle$, which belongs to the $L_{a_2 b_2}$ class in Ref. [27] for $a = 1$ and $b = 0$. This state is strictly semistable. Hence, there exists a sequence of operators $A_\alpha, B_\alpha, \ldots \in \text{SL}(d,\mathbb{C})$ such that $\lim_{\alpha\to\infty} A_\alpha \otimes B_\alpha \otimes \ldots |\chi\rangle \propto |\Psi_c\rangle$ for some critical state $|\Psi_c\rangle$ with non-vanishing proportionality factor, but no critical state is inside the SLOCC class of $|\chi\rangle$. To see that $|\chi\rangle$ is strictly semistable, note that

$$\lim_{\alpha\to\infty} \begin{pmatrix} e^{-\alpha} & 0 \\ 0 & e^{\alpha} \end{pmatrix} \otimes \mathbb{1} \otimes \begin{pmatrix} e^{\alpha} & 0 \\ 0 & e^{-\alpha} \end{pmatrix} \otimes \mathbb{1} \; |\chi\rangle = |\text{GHZ}\rangle \;, \tag{122}$$

and it may be easily verified that $|\chi\rangle$ and $|\text{GHZ}\rangle$ are not SLOCC equivalent. Similarly, it can be easily verified that both $|\chi\rangle|\text{GHZ}\rangle$ and $|\chi\rangle|\chi\rangle$ are strictly semistable with $|\text{GHZ}\rangle^{\otimes 2}$ being the critical state within the closure of their respective SLOCC class. Thus, the considered transformation is indeed a transformation from a strictly semistable and a polystable state to two strictly semistable states.

Finally, we find that two states from the null-cone may be transformed into one state in the null-cone as well as one strictly semistable state. With similar methods as above, it can be shown that the transformation involving the states

$$|\psi\rangle = |\chi\rangle^{\otimes 2} \tag{123}$$

$$|\bar{\psi}\rangle = |\text{W}\rangle^{\otimes 2} \tag{124}$$

$$|\phi\rangle = |\bar{\phi}\rangle = |\chi\rangle|\text{W}\rangle \;, \tag{125}$$

is indeed an example of that.

Let us remark here that, obviously, all of the considered transformations are also possible in reverse direction. We depict the considered examples in Figure 8. Note, though, that the orbit type may not be changed arbitrarily. For instance, it is impossible to transform two states in the null-cone to two polystable states. The reason for this is that, necessarily, the orbit

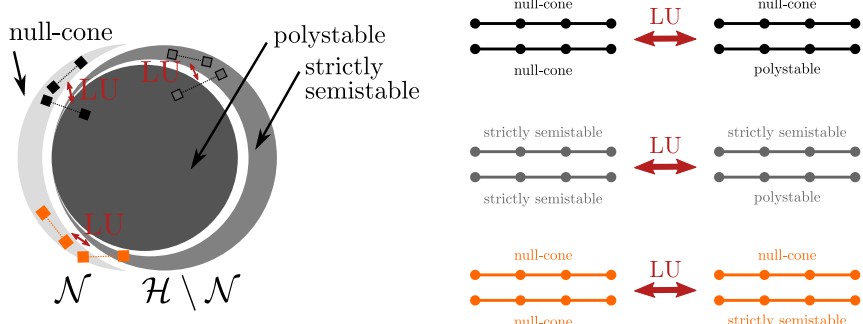

Figure 8: Illustration of how LUs operating jointly on two multipartite states may not only alter the SLOCC classes of the two considered states, but also the orbit type of the SLOCC classes. On the left we depict the Hilbert space partitioning into the three distinct orbit types: The null-cone $\mathcal{N}$, strictly semistable classes, and polystable classes. On the right hand side, we depict possible orbit type changes. First, we depict two states in the null-cone which can be transformed to one state in the null-cone as well as one polystable state. Then, we depict two strictly semistable states which can be transformed to one strictly semistable states as well as one polystable state. On the bottom, we depict two states in the null-cone that can be converted to a single state in the null-cone as well as a strictly semistable state. Concrete examples for all of the depicted scenarios are given in the main text. Finally, we indicate the considered pairs of states in the schematic picture of the Hilbert space on the left-hand side.

types of the tensor products of the states on both sites of the considered transformation must coincide. However, the tensor product of two states in the null-cone is in the null-cone, while the tensor product of two polystable states is polystable. More generally, it is impossible to transform any state in the null-cone together with an arbitrary state into two states neither of which is in the null-cone.

Finally, let us remark that it is not immediately clear whether the tensor product of a strictly semistable state and a polystable state (the tensor product of two strictly semistable states) is always strictly semistable, or may also be polystable. As we show in a following observation, the latter case would demonstrate reversible SLOCC catalysis. By this we mean it would provide an instance of three states $|\psi\rangle$, $|\bar{\psi}\rangle$, and $|\phi\rangle$ with the following properties. The states $|\psi\rangle$ and $|\phi\rangle$ are fully-entangled states of a Hilbert space with fixed local dimensions, which are not SLOCC equivalent. However, the states $|\psi\rangle|\bar{\psi}\rangle$ and $|\phi\rangle|\bar{\psi}\rangle$ are SLOCC equivalent. Irreversible SLOCC catalysis, i.e., an instance where $|\phi\rangle$ is lower-dimensional than $|\psi\rangle$, has been demonstrated in [40]. Regarding reversible SLOCC catalysis we make the following observation.

**Observation 11.** *An instance of two states $|\psi\rangle$ and $|\bar{\psi}\rangle$, one strictly semistable and one polystable, such that $|\psi\rangle|\bar{\psi}\rangle$ is polystable would provide an instance of reversible SLOCC catalysis.*

*Proof.* Consider a strictly semistable state $|\psi\rangle$ and a polystable state $|\bar{\psi}\rangle$ and suppose that the joint state $|\psi\rangle|\bar{\psi}\rangle$ is polystable. Let us denote the critical state that is in the closure of the SLOCC class of $|\psi\rangle$ by $|\phi\rangle$. Due to arguments used earlier in this appendix, $|\phi\rangle|\bar{\psi}\rangle$ is then in the closure of the SLOCC class of $|\psi\rangle|\bar{\psi}\rangle$. As we supposed that $|\psi\rangle|\bar{\psi}\rangle$ is polystable, its SLOCC class is closed and thus, $|\phi\rangle|\bar{\psi}\rangle$ is actually within the SLOCC class of $|\psi\rangle|\bar{\psi}\rangle$. However, $|\psi\rangle$ is not SLOCC equivalent to $|\phi\rangle$, as $|\psi\rangle$ is strictly semistable and $|\phi\rangle$ is critical. Note moreover that $|\psi\rangle|\bar{\psi}\rangle$ as well as $|\phi\rangle|\bar{\psi}\rangle$ are fully-entangled. The states $|\psi\rangle$, $|\phi\rangle$, and $|\bar{\psi}\rangle$ thus demonstrate reversible SLOCC catalysis. □

# B  Symmetries of $|\mathrm{GHZ}_d^n\rangle$

In this appendix, we prove Lemma 3 from the main text, which characterizes the local symmetries of generalized GHZ states. We recall the lemma here for completeness.

**Lemma 3.** *A local invertible operator is a symmetry of the state $|\mathrm{GHZ}_d^n\rangle$ (with $d \geq 2$ and $n \geq 3$) if and only if it can be written*

$$S = \left[ D(\vec{\gamma}^{(1)}) \otimes \cdots \otimes D(\vec{\gamma}^{(n)}) \right] X_\sigma^{\otimes n} , \tag{126}$$

*where*

$$D(\vec{\gamma}^{(i)}) = \mathrm{diag}(\gamma_1^{(i)}, \gamma_2^{(i)}, \ldots, \gamma_d^{(i)}) , \tag{127}$$

$$\gamma_j^{(n)} = \left( \prod_{i=1}^{n-1} \gamma_j^{(i))} \right)^{-1} , \quad \forall j \in \{1, \ldots, d\} \tag{128}$$

$$X_\sigma = \sum_{k=0}^{d-1} |\sigma(k)\rangle \langle k| , \tag{129}$$

*with $\sigma \in S_d$ any permutation of $d$ elements, $\vec{\gamma}^{(i)} = (\gamma_1^{(i)}, \ldots, \gamma_d^{(i)}) \in \mathbb{C}^d$ for $i = 1, \ldots, n-1$.*

*Proof.* The proof of this theorem is a straightforward generalization of the proof of the symmetries of $|\mathrm{GHZ}_2^3\rangle$ presented in Ref. [26]. It can be easily verified that the symmetries given in the theorem are indeed symmetries of $|\mathrm{GHZ}_d^n\rangle$. Let us now show that all symmetries of $|\mathrm{GHZ}_d^n\rangle$ are necessarily of that form.

To this end, let us consider an arbitrary symmetry $S = S^{(1)} \otimes \cdots \otimes S^{(n)}$ and compare the projections of the states $|\mathrm{GHZ}_d^n\rangle$ and $S|\mathrm{GHZ}_d^n\rangle$ onto $\langle ij|_{1,2}$ for $i \neq j$. For the first projection, it is straightforward to see that

$$\langle ij|\mathrm{GHZ}_d^n\rangle = 0 . \tag{130}$$

Since the same result must hold for the second projection, we have:

$$0 = \langle ij|S^{(1)} \otimes \cdots \otimes S^{(n)}|\mathrm{GHZ}_d^n\rangle \tag{131}$$

$$= \sum_k S_{i,k}^{(1)} S_{j,k}^{(2)} |\underbrace{k \ldots k}_{n-2}\rangle , \tag{132}$$

where we used the fact that the operator $\mathbb{1} \otimes \mathbb{1} \otimes S^{(3)} \otimes \cdots \otimes S^{(n)}$ is invertible. As the states $|k \cdots k\rangle$ in the equation above are orthogonal vectors as long as $n > 2$, it follows that $S_{i,k}^{(1)} S_{j,k}^{(2)} = 0$ for all $k$ and for all $i \neq j$. Moreover, this condition has to hold for arbitrary pairs of parties. From these conditions it follows that each matrix $S^{(i)}$ can only have one non-vanishing entry per column. Moreover, the positions of the non-vanishing entries must coincide for all parties. Since the matrices have to be invertible, the non-vanishing entries must be distributed over all rows. Adding up these constraints, we see that the matrices $S^{(i)}$ must correspond to the same column permutation of diagonal matrices. We can therefore write the symmetries as

$$S = \left[ D(\vec{\gamma}^{(1)}) \otimes \cdots \otimes D(\vec{\gamma}^{(n)}) \right] X_\sigma^{\otimes n} , \tag{133}$$

for some $\sigma \in S_d$. By applying $S$ to $|\mathrm{GHZ}_d^n\rangle$, it can be easily verified that $\vec{\gamma}^{(n)}$ has to be chosen as in Eq. (128). This shows that all symmetries are of the form given in Eq. (126) and completes the proof.

$\square$

# C   Further probabilistic multi-state transformations

In Section 4.4, we demonstrated that the multi-state setting can provide an advantage in probabilistic transformations. We demonstrated this by taking a state $|\psi\rangle$ such that the stabilizer of $|\psi\rangle^{\otimes 2}$ consisted of only $\mathbb{1}$ and SWAP. We then considered probabilistic transformations from two copies of a state to two distinct states. In this appendix, we now show that the multi-state setting does not always provide an advantage. To see this, we now consider the reverse of the previous example: i.e. transforming from two distinct states to two copies.

Let $|\psi\rangle$ be a normalized state such that the stabilizer of $|\psi\rangle^{\otimes 2}$ is unitary (note that this is a generalisation of the previous example, which required the stabilizer to consist of only $\mathbb{1}$ and SWAP). Then, by Eq. (7), we have:

$$p_{\max}^{SEP}\big(g_1 |\psi\rangle \otimes g_2 |\psi\rangle \mapsto |\psi\rangle^{\otimes 2}\big) \tag{134}$$

$$= \lambda_{\min}\left[\frac{G_1 \otimes G_2}{\||g_1 \otimes g_2 |\psi\rangle^{\otimes 2}\|^2}\right] \tag{135}$$

$$= \lambda_{\min}\left[\frac{G_1}{\|g_1 |\psi\rangle\|^2}\right] \lambda_{\min}\left[\frac{G_2}{\|g_2 |\psi\rangle\|^2}\right] \tag{136}$$

$$= p_{\max}^{SEP}\big(g_1 |\psi\rangle \mapsto |\psi\rangle\big)\; p_{\max}^{SEP}\big(g_2 |\psi\rangle \mapsto |\psi\rangle\big)\,. \tag{137}$$

That is, if the tensor product of two copies of a seed state has a unitary stabilizer, the multi-state regime provides no advantage in reaching two copies of the seed state via SEP. As was shown in Ref. [28], $p_{\max}^{SEP}$ for the individual state transformations is achievable via LOCC. Therefore, $p_{\max}^{SEP}\big(g_1 |\psi\rangle \otimes g_2 |\psi\rangle \mapsto |\psi\rangle^{\otimes 2}\big)$ is also achievable with LOCC, and the multi-state regime provides no advantage for this transformation.

# D   Further discussion of the application of Elementary and Power Sum Symmetric Polynomials

In Section 5.3.3, we used the elementary symmetric polynomials to prove that there are no non-trivial transformations in the case $d_\mu = d_\lambda = 2, 3$. To do this, we used the fact that the elementary symmetric polynomials provide necessary and sufficient conditions for tuples of variables to be equivalent up to reordering. In this appendix, we discuss elementary (and other fundamental) symmetric polynomials in more depth, and present the proof of Theorem 10.

To begin, let $x = (x_1, ..., x_n)$ be a tuple of $n$ variables. In addition to elementary symmetric polynomials, $e_k(x)$ (see Eq. (57)), we also have the power sum symmetric polynomials:

$$\psi_k = \sum_{i=1}^{n} x_i^k, \qquad \forall k \in \mathbb{N}\,, \tag{138}$$

again with $\psi_0 = 1$. These two families of symmetric polynomials are related by Newton's identities [61]:

$$k e_k(x) = \sum_{i=1}^{k} (-1)^{i-1} e_{k-i}(x)\psi_i(x), \quad \forall k \in \{1, \ldots, n\}\,. \tag{139}$$

As a consequence, the power symmetric polynomials give necessary and sufficient conditions for two tuples of real numbers to be equal up to reordering and, therefore, they also give

necessary and sufficient conditions for the existence of LU transformations. That is for two $d$-dim bipartite states, $|\lambda\rangle$ and $|\mu\rangle$, with Schmidt coefficients $\lambda = (\lambda_1, ..., \lambda_d)$ and $\mu = (\mu_1, ..., \mu_d)$, $|\lambda\rangle$ can be transformed with LUs into $|\mu\rangle$, if and only if

$$e_i(\lambda) = e_i(\mu), \ \forall i \in \{1, \ldots, d\}, \tag{140}$$

which is equivalent to

$$\psi_i(\lambda) = \psi_i(\mu), \ \forall i \in \{1, \ldots, d\}. \tag{141}$$

Finally, observe that the power symmetric polynomials are multiplicative under tensor product. That is,

$$\psi_i(x \otimes y) = \psi_i(x)\psi_i(y), \quad \forall i \in \mathbb{N}. \tag{142}$$

Therefore (as presented in Ref. [42]), by Newton's identities, the elementary polynomials over tensor products, $e_i(\lambda \otimes \mu)$ can also be expressed in terms of $s_i \equiv e_i(\lambda)$ and $t_i \equiv e_i(\mu)$. We can also easily deduce from this property:

$$|\lambda\rangle \rightarrow_{LU} |\mu\rangle \iff |\lambda\rangle^{\otimes n} \rightarrow_{LU} |\mu\rangle^{\otimes n}, \tag{143}$$

as this follows directly from Eq. (141) and Eq. (142).

We emphasize again that bipartite states are completely determined by the ESPs over their Schmidt coefficients. Therefore, if we let $\underline{s} = (s_2, s_3, ..., s_d)$ and likewise for $\underline{t}, \underline{\bar{s}}, \underline{\bar{t}}$, then the transformation

$$|\lambda\rangle \otimes |\mu\rangle \rightarrow_{LU} |\bar{\lambda}\rangle \otimes |\bar{\mu}\rangle, \tag{144}$$

is trivial (i.e. it corresponds to either $\mathbb{1}$ or SWAP) iff $(\underline{s}, \underline{t})$ is equal to $(\underline{\bar{s}}, \underline{\bar{t}})$ up to reordering. We now proceed to give necessary and sufficient conditions for this and prove Theorem 10:

**Theorem 10** (Equivalence between two tuples of tuples). *Let $\underline{s} = (s_1, s_2, ..., s_d) \in \mathbb{R}_+^d$ with $s_{i+1} \leq s_i$ and let $\underline{t}, \underline{\bar{s}}, \underline{\bar{t}}$ be defined similarly. Then $(\underline{s}, \underline{t})$ is equal to $(\underline{\bar{s}}, \underline{\bar{t}})$ up to reordering iff the following conditions hold.*

*1. $e_i(\underline{s}) + e_i(\underline{t}) = (\bar{"}), \ \forall i = 1..d$*

*2. $\sum_{i+j=k} e_i(\underline{s})e_j(\underline{t}) = (\bar{"}), \ \forall k = 1...2d$*

*where $(\bar{"})$ indicates the same as the LHS but with all variables barred.*

*Proof.* Consider the multivariate polynomials over $\lambda$ and $\mu$, with real parameters $s_1, \ldots, s_n, t_1, \ldots, t_n$:

$$p = p(\lambda, \mu; s_1, ..., s_n, t_1, ..., t_n)$$
$$= \left(\lambda - \prod_i (\mu - s_i)\right)\left(\lambda - \prod_j (\mu - t_j)\right), \tag{145}$$

and

$$\bar{p} = p(\lambda, \mu; \bar{s}_1, ..., \bar{s}_n, \bar{t}_1, ..., \bar{t}_n). \tag{146}$$

Then we have $p = \bar{p}$ if and only iff either

$$\prod_i (\mu - s_i) = \prod_i (\mu - \bar{s}_i), \text{ and } \prod_i (\mu - t_i) = \prod_i (\mu - \bar{t}_i), \tag{147}$$

or

$$\prod_i (\mu - s_i) = \prod_i (\mu - \bar{t}_i), \text{ and } \prod_i (\mu - t_i) = \prod_i (\mu - \bar{s}_i). \tag{148}$$

Let $\underline{s} = (s_1, s_2, ..., s_d) \in \mathbb{R}^d_+$ with $s_{i+1} \leq s_i$ and let $\underline{t}, \bar{\underline{s}}, \bar{\underline{t}}$ be defined similarly. Then we have either $\underline{s}$ is equal to $\bar{\underline{s}}$ up to reordering and $\underline{t}$ is equal to $\bar{\underline{t}}$ up to reordering or vice versa. As the tuples are ordered, this in fact holds only if they are actually equal. Which is to say, $p = \bar{p}$ iff $(\underline{s}, \underline{t})$ is equal to $(\bar{\underline{s}}, \bar{\underline{t}})$ up to reordering.

Expanding $p = \bar{p}$, we have:

$$p = \left(\lambda - \prod_i (\mu - s_i)\right)\left(\lambda - \prod_j (\mu - t_j)\right) \tag{149}$$

$$= \lambda^2 - \left(\prod_i (\mu - s_i) + \prod_j (\mu - t_j)\right)\lambda + \left(\prod_i (\mu - s_i)\right)\left(\prod_j (\mu - t_j)\right) \tag{150}$$

$$= \lambda^2 - \left(\sum_{i=0}^d (-1)^i e_i(\underline{s})\mu^i + \sum_{j=0}^d (-1)^j e_j(\underline{t})\mu^j\right)\lambda + \left(\sum_{i=0}^d (-1)^i e_i(\underline{s})\mu^i\right)\left(\sum_{j=0}^d (-1)^j e_j(\underline{t})\mu^j\right) \tag{151}$$

$$= \lambda^2 - \sum_{i=0}^d (-1)^i \left(e_i(\underline{s}) + e_i(\underline{t})\right)\mu^i\lambda + \sum_{i,j=0}^d (-1)^{i+j} e_i(\underline{s})e_j(\underline{t})\mu^{i+j} \tag{152}$$

$$= \bar{p} . \tag{153}$$

Comparing coefficients of $\lambda^n \mu^m$, we can deduce $p = \bar{p}$ iff the following conditions hold:

1. $e_i(\underline{s}) + e_i(\underline{t}) = (\bar{"})$, $\forall i \in 1..d$

2. $\sum_{i+j=k} e_i(\underline{s})e_j(\underline{t}) = (\bar{"})$, $\forall k \in 1...2d$

$\square$

As for any normalised bipartite state, $|\lambda\rangle$, $e_{i+1}(\lambda) < e_i(\lambda)$, we can use Theorem 10 to provide necessary and sufficient conditions for only trivial solutions to be possible, as explained in the main text.

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
