# Peer review of "Local Transformations of Multiple Multipartite States"

_SciPost Physics, doi:SciPost Phys. 11, 042 (2021)_

## Round 1 · Referee Report · Anonymous (Referee 1) · 2021-1-24

Report

In this paper, the authors study multi-copy transformations between pure quantum states. In detail, they consider LOCC protocols acting on several copies, and present various results (for the bipartite and multipartite case) about conditions under which states can be transformed into each other on a multi-copy level.

The paper is well written and comprehensive. At the moment, however, I am not entirely sure what long term impact can be expected from this work (see the acceptance criteria of SciPost and also below). Therefore, I would like to ask the authors to revise their paper.

I have the following remarks:

  • concerning the motivation: The authors motivate their study with potential applications for condensed matter systems as well as with the usage of quantum memories. In both cases, however, mixed states will inevitably play a role. So, the authors should comment on the question what can be learned by their classification for mixed states? For instance, assume that all their classification problems for two copies were solved, what does it imply for the mixed state case? This is an essential question to judge the long-term impact of this work.

  • even if one restricts the attention to pure states: What are the interesting problems for further studies and what are the long term applications? The conclusion does not answer this.

  • the readability would be improved if Figure 1 appears already in the introduction.

  • in the introduction, the authors write that "Therefore, all entangled bipartite states form a single SLOCC equivalence class". This seems not correct to me, as entangled states with a different Schmidt rank are not equivalent under SLOCC. The authors should be more precise here, and also later (e.g. in the first sentence of Section V).

  • at first sight, for the problem stated in Eq. 56 seems there is a trivial example, namely the following: If one takes as \mu two copies of a product state, and as \nu two copies of a Bell state, then one can argue as in Figure 2, and one finds an instance where the conditions of Eq. 56 hold. Is this reasoning correct? If yes, then maybe the authors should not claim that situations as in Eq. 56 are "surprising".

  • the permutations studied in the bipartite case (e.g., Figure 3) seem to be directly related to permutations that have been characterized in the context of high-dimensional entanglement (see Eq. 8 in Kraft et al., PRL 120, 060502 (2018)). Can the authors comment on that?

  • validity: top
  • significance: good
  • originality: high
  • clarity: top
  • formatting: excellent
  • grammar: excellent

Author:  Antoine Neven  on 2021-02-08  [id 1212]

(in reply to Report 1 on 2021-01-24)

We are grateful to the referee for their time and useful feedback. Taking their remarks into account (which we copy here for completeness), we made the following modifications to our submission.

concerning the motivation: The authors motivate their study with potential applications for condensed matter systems as well as with the usage of quantum memories. In both cases, however, mixed states will inevitably play a role. So, the authors should comment on the question what can be learned by their classification for mixed states? For instance, assume that all their classification problems for two copies were solved, what does it imply for the mixed state case? This is an essential question to judge the long-term impact of this work.

We naturally agree with the observation that the states one has to deal with in experiments are mixed but believe that it is nevertheless important to study pure state transformations for two reasons. First, from a theoretical point of view, understanding pure state transformations is a necessary step towards the study of mixed state transformations. Second, for experimental states that are "close" enough to pure states, the pure state transformations hold up to a certain fidelity of the final states. We added a paragraph summarizing the remark of the referee, as well as the two arguments given above, in the introduction of our paper.

even if one restricts the attention to pure states: What are the interesting problems for further studies and what are the long term applications? The conclusion does not answer this.

We agree that the impact of our results on further studies could be made clearer. We therefore modified our conclusions to stress that our results indicate that the structure of multi-state LOCC transformations is so involved (e.g. with the possibility of changing the SLOCC class of some of the states) that a complete characterization is unlikely. We also added that further studies should therefore focus on the asymptotic case, or on physically relevant sets of states (for which the multi-state LOCC structure might be simpler).

the readability would be improved if Figure 1 appears already in the introduction.

We followed this suggestion and moved Figure 1 to the introduction.

in the introduction, the authors write that "Therefore, all entangled bipartite states form a single SLOCC equivalence class". This seems not correct to me, as entangled states with a different Schmidt rank are not equivalent under SLOCC. The authors should be more precise here, and also later (e.g. in the first sentence of Section V).

We agree that this statement should be more precisely stated. We added in the introduction that we consider states with the same local ranks. We believe that it is not necessary to emphasize this again in section V, as the statement there explicitly refers to fully-entangled states, i.e. states for which all the single-party reduced density matrices have full rank.

at first sight, for the problem stated in Eq. 56 seems there is a trivial example, namely the following: If one takes as \mu two copies of a product state, and as \nu two copies of a Bell state, then one can argue as in Figure 2, and one finds an instance where the conditions of Eq. 56 hold. Is this reasoning correct? If yes, then maybe the authors should not claim that situations as in Eq. 56 are "surprising".

We would like to mention that the suggested transformation does not satisfy our assumptions. Indeed, for the bipartite transformations, we explicitly require that the dimensions of the initial states should be preserved through the transformation in order to avoid trivial transformations (see the first paragraph of Section V, with the example of the trivial $|\phi^+\rangle$ states transformation, which is similar to that suggested by the referee). More generally, we agree that it is not surprising to find sub-SWAP transformations (i.e. transformations as illustrated in Figure 2) also in the bipartite case. We nevertheless find more surprising that when the two initial states have coprime dimensions (in which case no sub-SWAP can be performed), many non-trivial solutions requiring various types of algebraic constraints on the Schmidt coefficients can still be found.

the permutations studied in the bipartite case (e.g., Figure 3) seem to be directly related to permutations that have been characterized in the context of high-dimensional entanglement (see Eq. 8 in Kraft et al., PRL 120, 060502 (2018)). Can the authors comment on that?

We thank the referee for pointing out this reference, where the authors study the decomposability of high-dimensional states into tensor products, as it is indeed related to the problem that we study in Section V. However, to characterize multi-state LU transformations of bipartite states, we have to look for all possible decompositions of a decomposable state and thus cannot exploit the algorithm proposed in [Kraft et al., PRL 120, 060502 (2018)], which decides whether a state is decomposable or not. We added a reference to this paper, as a related work, but also stressed the difference with our problem as we did above.

We hope this reply provides the requested clarification and would like to thank the referee once again, as we think their comments helped improving our paper.

Sincerely,

The authors

---

## Round 2 · Referee Report · Anonymous · 2021-5-14

Strengths
1-The manuscript opens a new window on multi-partite entanglement by using the notion of multi-state LOCC.
2-The authors develop a solid intuition how this notion behaves and answer a couple of key questions that clarify what multi-state LOCC can achieve.
Weaknesses
1-The manuscript only focuses on pure states (rather than also considering mixed states), but the authors make a compelling case that this already represents a meaningful first step.
2-The manuscript is non-exhaustive, as it rather explores a number of natural questions (in the context of multi-state LOCC) without the claim of being complete, as many questions are notoriously difficult (such as a full classification multi-partite multi-state transformations).
Report
The authors explore the notion of multi-state LOCC to study multi-partite entanglement. More precisely, they ask the question what types of pure states can be transformed into each other under regular LOCC if one is allowed to tensor an additional auxiliary state (in the same Hilbert space with the same multi-partite splitting) to the original state, such that the auxiliary state may transform into a different state (though still in a tensor product with the rest).
The main results are discussed in section IV and V. While section IV focuses on the multi-partite case, section V restricts to the bi-partite case (where the authors restrict to LU transformations). In particular, the authors show that multi-state LOCC can change the SLOCC class of a target state by only applying multi-state local unitaries and they identify several interesting multi-state local unitaries in the bi-partite setting. Overall, they make a compelling case that multi-state LOCC gives rise to rich structure that should be further explored.
The manuscript is well-written and makes interesting progress in the challenging field of multi-partite entanglement. The article thereby meets the SciPost selection criterion of "opening a new pathway in an existing or a new research direction, with clear potential for multipronged follow-up work". I therefore recommend publication in SciPost in the current version, but added a list of suggested changes, which are mostly cosmetic. However, I also tried to highlight a few spots where small modifications would potentially increase clarity - in particular, for some of the figures.
Requested changes
1-Shouldn't we have in equation (1) $S\in\mathrm{GL}(d,\mathbb{C})^{\otimes d}$? Or am I missing something?
2-There is a new paragraph after equation (3), i.e., see indent, which should probably be removed.
3-In equation (7) max is upright, while min is italic (variables).
4-It took me some time to fully understand the example illustrated in figure 2. In particular, I first thought that the illustration with the beige boxes has something to do with the three parties, before realizing that you (probably) just meant to indicate where the LU acts to swap the respective basis vectors. This could be just said more explicitly in the caption. It would have also helped me if you had maybe illustrated by arrows which respective basis states are permuted/swapped.
5-In lemma 3, there seem to be again unwanted indents / new paragraphs and in equation (20) there is an additional closing parenthesis in $\gamma^{(i))}_j$.
6-Figure 4 and the following figures look a bit pixeled and it would be good to convert it to vector graphics.
7-The authors could give a broader outlook on which questions remain open. They already did this in response to the first referee report (asymptotic case, restriction to physically relevant states), but it would also be interesting in regards to their results on the bi-partite case.
8-The label LU for the left-right arrows in the left figure are difficult to read. I would probably just remove them (and only have the arrows) and potentially make the state illustrations a bit larger
Author: David Kenworthy Gunn on 2021-06-11 [id 1500]
(in reply to Report 2 on 2021-05-14)We would like to thank the referee for their time and for their constructive feedback. Regarding the revisions suggested, we have the following responses:
It should be S∈GL(d,C)⊗n. We have also corrected a similar typographical error in Eq.2
We corrected this typographical error.
We corrected this typographical error and ensured all subsequent min/max are consistent.
We opted to add a further explanation in the caption. Hopefully, it is now clear the beige boxes indicate the local unitaries which are swapping the internal GHZ and W states.
We corrected this typographical error.
We followed this suggestion. Now, Fig. 4 and all subsequent images are vector graphics.
We took this suggestion on board. As a result, we modified our conclusion section. Firstly, we restructured the section so that the final paragraph is focused on the overall outlook of multi-state LOCC. As a result, in the final paragraph, we first discuss what we believe the focus of further investigations should be in the multipartite case (this discussion was moved from an earlier part of the conclusion). We then also provided further outlook with regards to the bipartite case (being mindful of the extensive research that has already been conducted, for example in entanglement catalysis).
In order to increase the readability of the label "LU" within Figure 8, we have increased the font size and we have chosen a different colour. Moreover, we have increased the size of the state illustrations a little bit, as suggested by the referee.
We would once again like to thank the referee for the detailed, constructive comments, and believe our paper has been improved because of them.
Sincerely, The authors.

---

## Round 2 · List of Changes

- The introduction was modified in response to the referee’s comments about the reason for studying pure state transformations. Additionally, we made small changes to make the text more readable and precise.
- In Section V, we corrected the typo in Fig.3 and added a sentence discussing permutations of Schmidt coefficients, providing a reference (Ref. [57]) to related work and an upper bound on the number of possible permutations (Eq. 57).
- We modified the conclusion to clarify the outlook of our work as suggested by the referee.
- We updated the Acknowledgements to include Ref.[55].

---

## Round 3 · List of Changes

- Some typographical errors have been corrected.
- Some figures have been edited to make them more understandable.
- The conclusion was restructured and modified to present a broader outlook for multi-state LOCC.

---

## Editorial Decision

published